# The impact of preferential trade agreements on bilateral trade: A structural gravity model analysis

**Xing Yao[1], Yongzhong Zhang[1]\*, Rizwana Yasmeen[2], Zhen Cai[1]**

**1** School of International Business, Southwestern University of Finance and Economics, Chengdu, China,
**2** School of Economics and Management, Panzhihua University, Panzhihua, Sichuan, China

\* yz_zhangjerry@126.com

## Abstract

Trade agreements are thought to raise trade integration, but existing preferential trade agreements (PTAs) are insufficient in measuring market access of products. This study develops a product-based coverage index of PTAs using the World Trade Organization (WTO) preferential trade agreements and calculates bilateral trade measures using the EORA multi-regional input-output (MRIO) tables covering 189 countries worldwide over the period 1990–2015; the structural gravity model is employed to test how PTAs affect bilateral trade. Our findings show that countries sharing a common PTA could boost the trade volume compared to those without PTAs, supporting the trade creation effect. However, the trade promotion effect of the product-based coverage index of PTAs is significant only if the member countries are low-and middle-income countries. Further, the wide range of product liberalization brought by PTAs can promote global production networks by stimulating the trade of intermediate goods. Our results are important for understanding the market access effect of PTAs with the increasing development of trade integration and global value chains (GVCs).

## 1. Introduction

Since the 1990s, the delay in the Doha Round negotiations caused the multilateral trading system to weaken, and countries turned to preferential trade agreements (PTAs) negotiations as a solution. RTAs are reciprocal preferential trade agreements between two or more partners, which belongs to PTAs. Although WTO website denotes as RTAs, we clarify it as PTAs. Thus, a new wave of regional economic integration has emerged [1, 2]. With the weakening role of the World Trade Organization (WTO) multilateral trading system, the globalization pattern is moving toward regional economic integration. The number of PTAs has increased over the last two decades [3, 4]. For example, the South Asian Association for Regional Cooperation (SAARC) member formed the Bay of Bengal Initiative for Multi-Sectoral Technical and Economic Cooperation (BIMSTEC) in 2004 to promote economic integration. In 2006, SAARC members formed their PTAs (SAFTA—South Asian Free Trade Area) and ASEAN

**Data Availability Statement:** Data used in this study came from multiple third party sources and can be accessed via the resources below: All Regional Trade Agreements (RTAs) DATA files are available from the World Trade Organization

Regional Trade Agreements Database: http://rtais.wto.org/UI/PublicMaintainRTAHome.aspx. All The depth and flexibility variable data files are available from the Design of Trade Agreements (DESTA) Database: https://www.designoftradeagreements.org/downloads/. All Trade data files are available from The Eora Global Supply Chain Database: https://www.worldmrio.com/. All Geo Dist data files such as bilateral distance, area, common language and colonies come from the CEPII GeoDist database: http://www.cepii.fr/anglaisgraph/bdd/distances.htm. All Gross Domestic Product and population data files are available from the World Development Indicators: https://databank.worldbank.org/source/world-development-indicators.

**Funding:** This work was supported by the National Science Foundation of China (No. 71903157 and 72003152), Ministry of Education Project of Humanities and Social Sciences (No. 19XJC790013). The funders had no role in study design, data collection and analysis, decision to publish, or preparation of the manuscript.

**Competing interests:** The authors have declared that no competing interests exist.

(Association of Southeast Asian Nations) member countries established the ASEAN Free Trade Area (AFTA) in 1992 to boost trade among member countries. According to WTO statistics, around 82 different PTAs were in force globally before 2000. In contrast, three times more PTAs have been in force in the last 20 years (Fig 1). Although there is a difference between the cumulative notifications of PTAs and the cumulative PTAs in force, the common increasing trend proves the wave of regional economic integration. Meanwhile, the PTAs can be divided into new PTAs and accession to an existing PTA, Fig 1 shows that although accession to existing PTAs occurs in certain years, signing new PTAs takes up a larger proportion. In terms of different types of trade, the ratio of service notifications has gradually increased since the 21st century.

Countries are permitted to enter into PTAs under specific conditions covering trade in goods (Article XXIV of the General Agreement on Tariffs and Trade 1994), regional or global arrangements for trade in goods between developing country members (Enabling Clause), and agreements covering trade in services (Article V of the General Agreement on Trade in Services). PTAs are an exception to WTO's non-discrimination principle, because only their signatories enjoy more favorable market access conditions. Thus, different PTAs are heterogeneous in terms of the market access level and the cooperation beyond tariff reductions, in services trade, investments, standards, public procurement, competition and intellectual property rights [2–4]. Previous studies have constructed variables to measure the heterogeneity of PTAs according to the content of PTA documents [5–7]. They construct depth variables to capture cooperation in services trade, investments, standards, public procurement, competition and intellectual property rights instead of product market liberalization of PTAs and flexibility variables to measure whether the PTAs allow states to withdraw concessions temporarily. Some scholars have tried to analyze schedules of commitments in services PTAs by looking at specific market access and national treatment commitments in the 155 sub-sectors of the Services

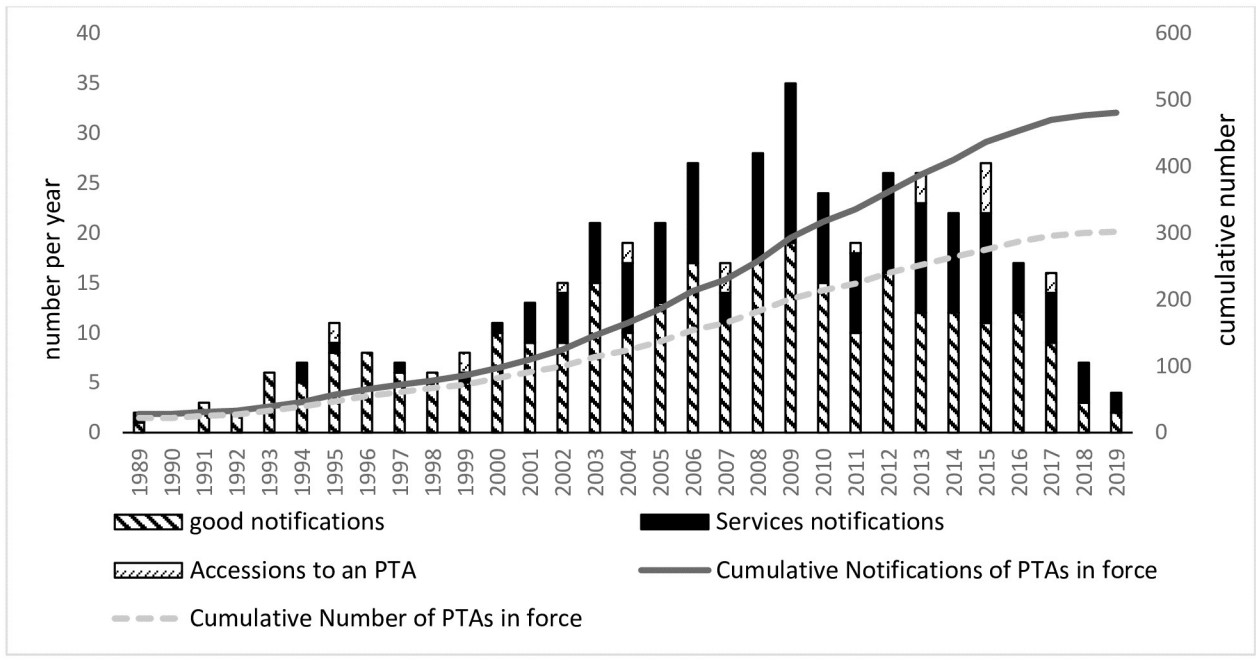

**Fig 1. Global trend of PTAs over the years of 1989–2019.** The y-axis on the left shows the number of different types (good notifications, services notifications and accessions to a PTA) of PTAs per year. The y-axis on the right shows the cumulative notifications and the cumulative number of PTAs in force. Source: WTO website RTA database.

Sectoral Classification List (MTN.GNS/W/120) [8–10]. However, few studies consider goods and services together and construct a product-based measurement to reveal the market liberalization level of PTAs. Global value chains (GVCs) play increasingly significant roles in the production of goods [11] and the shortage in the supply of some products under the COVID-19 pandemic [12], measuring the product coverage of PTAs and identifying the effect of the product-based market liberalization of PTAs on bilateral trade is important.

More scholars have studied how PTAs affect bilateral trade and conclude that PTAs could influence bilateral trade through trade creation and diversion effects [13–16]. Scholars established that the trade creation (increased trade because of relative efficiency) impact could be higher than the trade diversion effect (increased trade because of preference) [17, 18]. Some scholars believe that PTAs are not consistent with an important principle of the multinational trading system and lead to unfair trade practices in trade disciplines [19–24]. PTAs have become a very appealing aspect of the international trading system [25] and according to World Bank (2018) [26], PTAs are currently the central topic of many policymakers and scholars. PTAs pave the path to achieve "deep" economic integration by eliminating tariffs. PTAs are flexible, and the product-based coverage of PTAs may differ significantly. Most of the existing studies on the effect of PTAs on bilateral trade mostly use dummy variables to measure whether countries share PTAs [27–29]. A few scholars use the depth variable and found that the depth of PTAs matters for bilateral trade [5]. Scholars have investigated the effect of PTAs on intermediate trade from GVCs perspective and found that PTAs can facilitate supply chain activity. When decomposing bilateral trade into different components, they mainly focus on the average effect of PTAs on GVCs (using a dummy variable to measure PTAs) and the effect of provisions related to services, investment, and competition on GVCs [30–32]. Starting from the product-based coverage of PTAs and testing its effect on different types of trade is meaningful in the global production network. Furthermore, some scholars have tried to identify the heterogeneous effect of different PTAs on trade development and found that different PTAs can have different effects on bilateral trade [33]. However, most studies focus on the average effect of PTAs. Additionally, knowledge of the product market access level brought by PTAs on trade among different types of members is insufficient. In conclusion, constructing the product-based coverage of PTAs and investigate the effect of the product-based coverage of PTAs on different types of bilateral trade can fill the literature gap by showing how the product market access level affect bilateral trade.

In this study we downloaded each PTA document (302 in total) and extracted all the products whose trade barriers are removed to promote free trade. Spanish and French translations were used to identify all products mentioned in the PTAs. We then match the products mentioned in each PTA document with the International Standard Industrial Classification of All Economic Activities (ISIC Rev.4), which includes 56 industries (1 to 23 are agricultural and manufacturing goods industries, 24 to 56 are service industries) and calculate the product-based coverage index (*coverage index of PTAs*) based on the ratio of industries covered in PTAs. ISIC Rev. 4 is a standard classification of economic activities arranged to be classified according to their activity. The categories of ISIC at the most detailed level (classes) are delineated according to what is, in most countries, the customary combination of activities described in statistical units and considers the relative importance of the activities included in these classes. While ISIC Rev.4 continues to use criteria such as input, output, and use of the products produced, more emphasis has been given to the character of the production process in defining and delineating ISIC classes. Our measure is different from the depth and flexibility measure of PTAs proposed by existing literature, focusing on services trade, investments, standards, public procurement, competition and intellectual property rights. Further, we use the EORA input-output database to calculate bilateral trade volume and decompose total trade

into components in terms of trade types (intermediate goods, final goods, commodities and service trade). Based on the measures above, we then investigate the effect of the coverage index of PTAs on bilateral trade using the structural gravity model, which can better estimate the marginal effect of PTAs on bilateral trade and are popular among previous studies [28, 29, 33]. We further investigate the heterogeneous effect of PTAs on trade by considering the economic development level of members and different types of trade. We find that the coverage index of PTAs has a significantly positive effect on bilateral trade and the results are robust using different fixed effects. However, the trade promotion effect of the coverage index of PTAs is only significant if the member countries are low and middle-income countries, different from the depth and flexibility measures of PTAs. For different trade types from the GVCs perspective, we find that the coverage index of PTAs can promote all types of trade, consistent with the effect of the depth variable.

This study contributes to PTAs and bilateral trade literature in the following ways. In contrast to previous literature, we construct the product-based coverage index of PTAs using WTO regional trade agreement datasets to measure the level of market access of PTAs. Although some scholars have measured the depth and flexibility of PTAs, few have explored their heterogeneous effect using different measures. We constructed the coverage index of PTAs and added other measurements in the regression to explore the effect of market access on bilateral trade by controlling other variables. Furthermore, we divide countries into low-, middle-, and high-income groups and tested the effect of PTAs on trade among different groups, the results show that the coverage index of PTAs is only significant if the members are low-income and middle-income countries; for these countries, market access barriers are still there, and liberalization of more products in the PTAs can promote trade. Subsequently, we decompose the total trade into components and investigate the effect of the coverage of PTAs on different trade components. Specifically, this study uses the EORA input-output database to calculate the bilateral trade flow between s countries in agreements. The distinction between using the EORA database is that it considers the intermediate goods between national industries, and considers the trade volume of final consumption. We divide trade into intermediate, final goods, service and commodity trade goods to evaluate the impact of PTAs on them from the GVCs perspective.

## 2. Literature review

In this section, we review the related literature to clarify our contribution from the literature perspective. We mainly focus on two strands of literature: the effect of PTAs on bilateral trade and the design of PTAs.

PTAs have been proliferating for the last twenty years. A large body of literature has studied various aspects of this phenomenon and study its design. Some researchers have highlighted that bilateral trade agreements are obstacles to free trade [34]. Regarding the obstacle [35–39], researchers argued that PTAs undermine multilateral growth and unleash a protectionist spiral. They further stated that PTAs effects like the spaghetti bowl effect and reduced countries' incentive to enter multilateral trade agreements. Regionalism leads to welfare losses in both member and excluded countries [40]. In a Ministerial Meeting of the WTO in Doha (2001), governments expressed their opinion that "regional trade agreements can play an important role in promoting the liberalization and expansion of trade" [41]. The motivation for this evolution in PTAs is to ease tariff barriers and 'new age' matters such as foreign direct investment, services, labor, and environmental standards [28]. More scholars have applied the gravity model to estimate the effect of PTAs and find a positive effect on bilateral trade [39, 40]. However, most studies conceptualize PTAs as a dichotomous variable, namely whether countries

sign an agreement or not and hence treat PTAs as if they were all equal in purpose while esti-mating the effects of PTAs. The results are fruitful, and they conclude that, on average, PTAs can promote bilateral trade for member countries. Some scholars estimate that the long-run effect of PTAs on bilateral trade flows is 100% and the effect varies considerably across trade agreements [42, 43]. Others applied the structural gravity model to analyze the PTAs and con-cluded that the average treatment effect of PTAs on trade flows was 236% [44]. Dembatapitiya and Weerahewa [45] investigated the SAFTA, European Union (EU), ASEAN, BIMSTEC, and North American Free Trade Agreement (NAFTA) regional trade impact and found mixed results. For example, the co-efficient for the EU is significant, while SAFTA, ASEAN, BIM-STEC and NAFTA do not significantly impact bilateral trade. Sampson [46] studied on the evolution of China's PTAs and stated that the increasing network of China's PTAs is important and strategic for the Asian region. Pant and Paul [47] evaluated PTAs for India and argued that PTAs are good for the intra-regional trade volume and welfare of countries. Scholars also assess the ex-post trade effects by applying the gravity model [48, 49]. Carre're [50] studied ex-post PTAs and claimed that intra-regional trade mostly tied to a reduction in imports. How-ever, regional agreements back weak governments to implement reforms and stabilize them despite domestic opposition. Partners may learn the advantages of liberalization once they practice limited free trade.

Until recently, scholars investigated the content and design of PTAs [51] and studied regional specifications [52] or explained functional differences in design, for example, dispute settlement [53] and flexibility provisions [54]. Furthermore, they found that tariff cuts, and other market access and trade-related provisions in PTAs concerning topics such as invest-ments and intellectual property rights matter for trade flows [5], and that such deep agree-ments are usually flexible in adjusting their policies for other purposes and withdraw the PTAs without violating the terms of an agreement [6]. In addition to market access to goods, many PTAs today include provisions in trade disciplines such as services, investment, standards, intellectual property, and competition rules, as well as a host of issues not directly related to trading, such as the environment. At the beginning of the study of PTAs' design, scholars usu-ally focus on specific PTAs and measure the strength of a wide variety of provisions in the legal texts of PTAs [10, 55]. With increasing number of documents on PTAs available on the WTO website, scholars have extended their objects and started to pay greater attention to the scope and depth of these agreements [5–10, 55–58]. They focus on the broader economic integration rights in goods, services, and factor markets brought by PTAs. For example, Dür et al. [5] used two different measures to operationalize the depth of PTAs. The first measure of depth is an additive index that combines seven key provisions included in PTAs, and the second one relies on latent trait analysis. Baccini et al. [6] measured the flexibility of PTAs and found a positive relationship between depth and flexibility for PTAs. Hofmann et al. [7] offered a detailed assessment of preferential arrangement. They examined the coverage and legal enforceability of provisions regulating a large set of policy areas. In addition to the depth of PTAs, many scholars have studied the breadth of PTAs [59–61]. Miroudot et al. [10] attempted to construct an industry-based coverage index of PTAs; however, they only studied 56 services PTAs. Limão claims that a broader PTA is one where partners seek to increase market access in prod-uct markets, and in markets for capital, labor, and technology [59]. Based on this, Hofmann et al. construct the measure based on a count of the provisions covered by a PTA and can focus either on the 18 'core' provisions [60] and recent scholars also build a measure following this method [61]. Although some scholars have constructed variables to measure the heteroge-neity of PTAs, most of them focus on the non-trade aspects of PTAs, and few measure the cov-erage index from the product-based perspective, which is important in the context of global production network.

In addition to the effect of PTAs on bilateral trade and the design of PTAs, this study also investigates how PTAs affect intermediate goods trade. Existing literature referring to the GVCs tends to study how deep trade agreements affect different trade types [61–63]. Laget et al. [62] use trade data in both value-added and gross forms to provide a complete picture of GVCs trade as possible and found that the positive impact of deep trade agreements on GVCs integration is driven by value-added trade in intermediate rather than final goods and services. In contrast to Laget et al. [62], scholars [61, 63] use the EORA multi-regional input–output (MRIO) data to derive variables for trade in value-added, as these offer greater country coverage. They found that broader PTAs have a larger impact on trade flows involving intermediates relative to flows involving all products, suggesting that GVCs trade is particularly sensitive to the scope of trade policy cooperation [61]. However, they measure the coverage index of PTAs based on the number of provisions, and we still do not know whether the products mentioned in the PTAs promote cooperation in the global production network. With the development of global production network, a product-based measure for the coverage of PTAs is needed to study the heterogeneity of PTAs and how this affects bilateral trade structure.

## 3. Empirical model and data specification

### 3.1 The structural gravity model

The gravity model has been the most commonly used model to predict the effect of PTAs on bilateral trade and it has been evaluated in different ways (for details, see Feenstra 2015, Head and Mayer 2014) [64, 65]. However, the most recent version of the model is a structural gravity model, which is originated by Anderson (1979) [66], and make renowned by Eaton and Kortum (2002) [67] and Anderson van Wincoop (2003) [68]. Baier and Bergstrand (2007) further extended it to the panel dimension [69]. The derivation of the gravity model can be expressed as follows:

$$X_{ij} = \frac{A_i w_i^{-\theta} \tau_{ij}^{-\theta}}{\Sigma_l A_l w_l^{-\theta} \tau_{lj}^{-\theta}} E_j \qquad (1)$$

Suppose that, $X_{ij}$ is the value of exports from country i to country j. Here, $E_j$ is the total expenditure of the goods purchased in j (including goods produced domestically and internationally). The share of j country expenditure allocated specifically to the goods from country i is directly subject to the following three factors: (i) $A_i$ denotes to production technologies used in i; (ii) $w_i$ is the wage in the country i (3) $\tau_{ij}$ is the iceberg cost of products shipped from country i to j. We assume that goods from other countries are imperfectly substitutable. Therefore, the impact of production and trade costs on trade is subject to a constant trade elasticity $\theta > 1$.

Rewrite Eq (1) as:

$$X_{ij} = \frac{A_i w_i^{-\theta} \tau_{ij}^{-\theta}}{P_j^{-\theta}} E_j \qquad (2)$$

where $P_j^{-\theta} = \Sigma_l A_l w_l^{-\theta} \tau_{lj}^{-\theta}$. $P_j^{-\theta}$ represents the aggregate cost faced by importing countries [68].

It has been assumed that when the two countries are involved in PTAs, the trade cost ($\tau_{ij}^{-\theta}$) between i and j will decrease. Thus, we focus on the average effect of PTAs on bilateral trade as highlighted by Baier and Bergstrand [69]. By rewriting Eq (2) with time subscript t and an

error term as follows:

$$X_{ij,t} = \exp\left( \ln A_{i,t} w_{i,t}^{-\theta} + \ln \frac{E_{j,t}}{P_{j,t}^{-\theta}} + \ln \tau_{ij,t}^{-\theta} \right) + \varepsilon_{ij,t} \tag{3}$$

Considering study objective, we use the following generic functional form of the trade cost term:

$$\ln \tau_{ij,t}^{-\theta} = Z_{ij}\delta + \beta PTA_{ij,t} + u_{ij,t} \tag{4}$$

where $Z_{ij}$ is a set of time-invariant controls of trade costs between i and j, with coefficient vector δ, (for instance, bilateral distance and other historical factors). $Z_{ij}$ is the unobserved component correlated with PTAs and may produce biased estimates, Baier and Bergstrand (2007) [69] recommended pair-specific fixed effects instead of $Z_{ij}\delta$.

Our baseline specification for estimating the average effect of PTAs on trade barriers is as follows:

$$X_{ij,t} = \exp(\eta_{i,t} + \varphi_{j,t} + \gamma_{\bar{ij}} + \beta PTA_{ij,t}) + \varepsilon_{ij,t} \tag{5}$$

We rewrite Eq (5) and get our baseline regression model:

$$\begin{aligned} \ln X_{ij,t} &= \alpha_0 + \beta_1 PTA_{ij,t} + \beta_2 \ln GDP_{i,t} + \beta_3 \ln GDP_{j,t} + \beta_4 \ln Dis_{i,j} + \\ &\quad \beta_5 Contig_{i,j} + \beta_6 Comlang_{i,j} + \beta_7 Colony_{i,j} + \beta_8 Comcur_{i,j} + \varepsilon_{ij,t} \end{aligned} \tag{6}$$

where $X_{ij}$ is bilateral trade flow between agreements countries, *PTAs* is PTAs, $GDP_i$, denotes the economic scale of the importer countries, while $GDP_j$ is the economic scale of the exporter countries, *Dist* is the physical distance between country "i" and country "j", *Contig* indicates the adjacent countries, comlang signifies the common language, colony indicates colonial relation between regional trading partner country, and comcur signifies whether the two partner countries use the same currency.

Furthermore, the baseline model may have potential missing variables that leads to misleading results. Therefore, we add multiple fixed effects to the baseline model to control the potential missing variable and obtain the regression model (7).

$$\ln X_{ij,t} = \alpha_0 + \beta_1 PTA_{ij,t} + \gamma_{\bar{ij}} + \eta_{i,t} + \varphi_{j,t} + \varepsilon_{ij,t} \tag{7}$$

where $\eta_{i,t}$ and $\varphi_{j,t}$ denote country-level factors on the importer and exporter sides, respectively. $\gamma_{\bar{ij}}$ is a symmetric pairwise fixed effect that strips out all time-invariant determinants of trade barriers between i and j. In contrast to Baier et al. (2019) [70], we used both dummy variables to measure whether there are PTAs between countries and the openness of the PTAs, measured using the information extracted from the PTA documents.

## 3.2 Construction of the product-based coverage index of PTAs

We follow scholars who study the breadth of PTAs [10, 59–61] to construct a coverage index based on its content. We focus on the product-based coverage of PTAs instead of referring to other aspects. In contrast, the existing literature focuses on the industry-based coverage index of service PTAs [10] or a count of provisions covered by a PTA [59–61]. Based on WTO rules, PTAs are under specific conditions covering trade in goods (Article XXIV of the General Agreement on Tariffs and Trade 1994), regional or global arrangements for trade in goods between developing country members (Enabling Clause), and agreements covering trade in services (Article V of the General Agreement on Trade in Services). We first manually

downloaded all PTA documents (302 in total) from WTO website to extract all products whose trade barriers are removed to promote free trade. We extracted all products in goods and keywords in service of each PTA document, Spanish and French translators were used to identify all products mentioned in the PTAs. To calculate a comparable measure for all PTAs, we match the products mentioned in each PTA document with the ISIC Rev.4, which includes 56 industries (1 to 23 are agricultural and manufacturing goods industries, 24 to 56 are service industries). We identify whether the PTA covers this industry by searching all products in goods industries and keywords in service industries. For example, China and South Korea signed a PTA which was in force in late 2015. We searched the document which is in Chinese, and it mentioned products such are fish, wood, bank service, etc. Based on the ISIC Rev.4, we then matched fish to "3. Fishing and aquaculture", wood to "7. Manufacture of wood and products of wood and cork, except furniture; manufacture of articles of straw and plaiting materials", bank service to "43. Activities auxiliary to financial services and insurance activities" respectively. The coverage index of PTAs is calculated based on Eq (8), where $Covered_i = 1$ if the RTA document covers the products of industry "i", otherwise $Covered_i = 0$. This is similar to the count of provisions covered by a PTA [59–61], however, we differ from them by using the ratio of covered industries based on the products.

$$PTA\_ratio = \frac{\sum_{i=1}^{56} Covered_i}{56} \qquad (8)$$

We take China as an example to verify our product-based coverage index of PTAs, which is shown in Fig 2. We choose China because the design of related PTAs is shown on the government website, available publicly. (http://fta.mofcom.gov.cn/). China's regional trade agreements since 2000 show the characteristics of differentiation: the China-Switzerland free trade agreement and China-Iceland free trade agreement, which came into effect on 1 July, 2014, have a broad coverage in terms of trade in goods and trade in services. As important non-EU countries, Switzerland and Iceland are China's important economic and trade partners in Europe. The two agreements have wide coverage, high level of openness, and many preferential policies. They are high-quality and wide-ranging free trade agreements. They are also one of the highest level and most comprehensive free trade agreements reached by China in recent

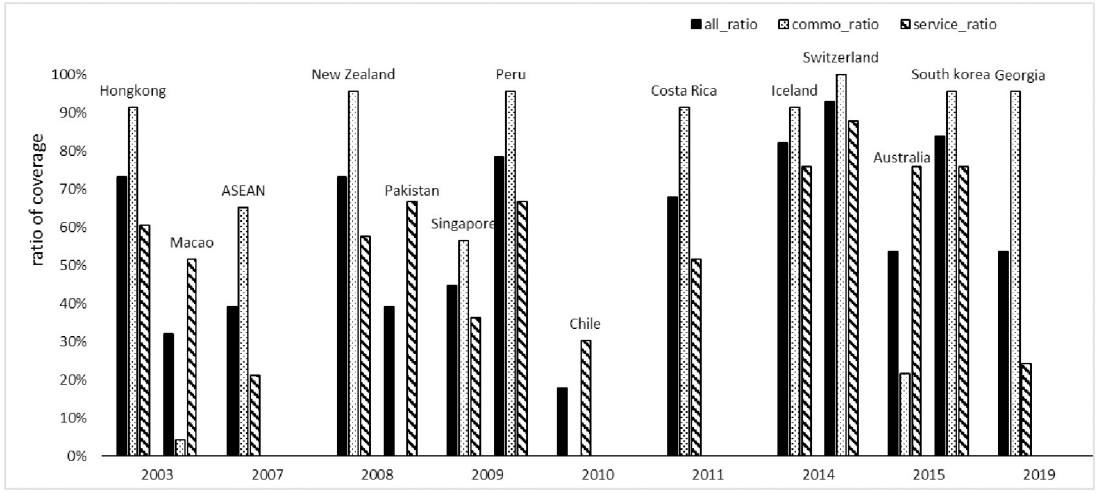

**Fig 2. The product-based coverage index of PTAs referring to China.**

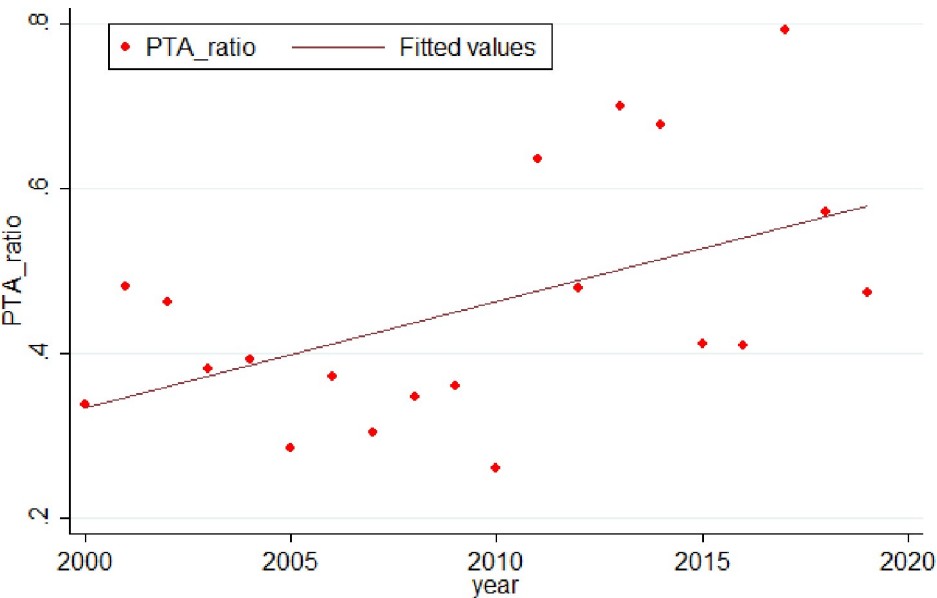

**Fig 3. The trend of coverage ratio index of PTAs over the years 2000 to 2019.** The documents of PTAs are from WTO website: http://rtais.wto.org/UI/PublicMaintainRTAHome.aspx. We calculate the mean value of coverage index of PTAs in each year and we can see that the trend is upward.

years. The product-based coverage index shows that the coverage rate of goods and service trade of China-Switzerland free trade agreement is 100% and nearly 90% respectively, which has a strong fit with the official data.

We then show the trend of the product-based coverage index of PTAs in Fig 3. It is evident from the Figure that product-based coverage index of PTAs has been increasing since the 21st century. However, it also shows that the variance of this trend is large. We calculated the mean value of the product-based coverage index of all PTAs based on the year in force, which could affect the trend due to the time lag. Additionally, when we look at the components, we find that the coverage index of service industries is growing more significantly than that of the goods industries. Overall, an upward trend can be concluded if we focus on the whole picture.

In the empirical regression, we construct country-pair observations. In multilateral agreements, a country and an economic organization that shares the same PTA are paired, for example, ASEAN. Any two countries in ASEAN countries share the same PTA and the coverage index of PTA. In 2002, China and ASEAN Free Trade Area signed by China and ASEAN countries (CAFTA). In this case, China and ASEAN countries share the same PTA and the coverage index of PTA. To conclude, if an agreement is signed within an economic organization, all countries included in the economic organization would be matched. Finally, multiple countries would share the agreement. Similarly, we treat other multilateral agreements and obtain the reciprocal industry penetration of agreement countries. Based on the coverage index of PTAs, this study examines the relationship between PTAs and bilateral trade.

### 3.3 Construction of bilateral trade variables and other variables

Following previous studies [61, 62], we use the latest release of the EORA MRIO tables [71]. The rows in an MRIO table indicate a gross output from a particular industry in a particular country and comprise two main components. First is intermediate use that provides information on domestic industries and industries in other countries. Second is the final demand

information, split between the demand for final goods from both domestic and foreign sources. The columns in the MRIO table provide information on the amounts of intermediates needed for the production of gross output. The column sum thus gives the sum of the domestic and foreign production of intermediates that are used in the production of output in a particular industry and country [61]. The EORA input–output tables have the distinct advantage of offering wide country coverage than other input-output databases, including the 189 economies over 1990–2015. We can calculate different total trade components based on the EORA MRIO tables, and investigate the effect of product-based coverage index of PTAs on bilateral trade.

Additionally, the gravity approach requires others variant and invariant variables. For example, GDP was used to measure the economic scale of countries obtained from the IMF datasets [72]. The geographical distance was extracted from the CEPII GeoDist database [73] to control for trade cost between countries. Data for official language were extracted from the Central Intelligence Agency's (CIA's) World Fact Book [74]. It is worth noting that in the specific regression, the fixed effect between the year fixed effect and the country pairing added according to the theoretical model to control the impact of other potential missing variables.

To verify our results, we also added other measures of PTAs and compared them with the product-based coverage index in our robustness checks. Specifically, we added the depth and flexibility variables in robustness checks to control for cooperation beyond tariff reductions, in services trade, investments, standards, public procurement, competition and intellectual property rights. We download the depth and flexibility variables from the website (https://www.designoftradeagreements.org/downloads/). Dür et al. [5] used two different measures to operationalize the depth of PTAs. The first measure of depth is an additive index that combines seven key provisions that can be included in PTAs, and the second one relies on latent trait analysis (depth_rasch) [5]. Baccini et al. [6] measured the flexibility (flexescape) of PTAs based on their content and found that positive relationship between depth and flexibility holds for PTAs. Our measure focuses on the market access of products brough by PTAs and we expect that the product-based coverage index of PTAs has a significantly positive effect on bilateral trade with other measures of PTAs controlled.

Descriptive statistics for the variables are presented in Table 1. The mean value of the product-based coverage index of PTAs and the PTA dummy suggests that 15.5% of the country pairs share a common PTA, while the average of the product-based coverage index of PTAs is 3.79%. All non-invariant gravity factors such as Contig, Comlang, Colony, and Comcur were normalized to range between 0 and 1. The maximum and minimum income scales are 30.49 and 18.63, respectively. The mean value of the geographical distance is approximately 8.704. The relationship between trade and PTAs shown in Fig 4. It is clear that PTAs and the coverage trade index have an upward trend. However, the bilateral trade increases comparatively more with the product-based coverage index of PTAs. This implies that PTAs decrease tariffs and promote trade between partner countries.

## 4. Empirical results and discussion

### 4.1 Baseline results of how PTAs affects bilateral trade

First, we test the effects of PTAs on bilateral trade. The results of the panel structural gravity model are presented in Table 2. We first used the dummy variable (PTAs) to measure whether a trade agreement exists the two-countries, and the results are shown in column (1)–(3). The results in column (1) of Table 2 shows that the impact of PTAs on bilateral trade is positive and significant. This implies that a trade agreement helps increase the trade flow between the member countries of PTAs. These results are consistent with the existing study [75]. We also

**Table 1. Description.**

| Variables | Definition | N | Mean | Sd | Min | Max |
|---|---|---|---|---|---|---|
| lntrade | *Total Trade value* | 721,149 | 7.699 | 2.892 | 4.268 | 24.86 |
| lnmid_trade | *Intermediate goods trade value* | 721,149 | 7.350 | 2.795 | 3.930 | 24.86 |
| lnfd_trade | *final goods trade value* | 721,149 | 6.250 | 3.153 | 2.488 | 23.55 |
| lnservice | *service trade value* | 721,149 | 7.083 | 2.984 | 3.574 | 24.72 |
| lncommo | *goods trade value* | 721,149 | 6.846 | 2.764 | 3.296 | 23.59 |
| PTA_ ratio | *coverage index of PTAs* | 721,149 | 0.0379 | 0.111 | 0 | 1 |
| PTA | *PTA Dummy* | 721,149 | 0.155 | 0.362 | 0 | 1 |
| depth_rasch | *Depth of PTAs* | 721,149 | -0.027 | 0.463 | -1.433 | 2.267 |
| flexescape | *Flexibility of PTAs* | 721,149 | 0.500 | 1.276 | 0 | 4 |
| lnYi/lnYj | *Log GDP* | 721,149 | 23.81 | 2.224 | 18.63 | 30.49 |
| lnDis | *Log Distance* | 721,149 | 8.704 | 0.809 | 0.632 | 9.892 |
| Contig | *Whether contig* | 721,149 | 0.0189 | 0.136 | 0 | 1 |
| Comlang_ | *Whether common-language* | 721,149 | 0.134 | 0.341 | 0 | 1 |
| Colony | *Whether colony* | 721,149 | 0.0124 | 0.111 | 0 | 1 |
| Comcur | *Whether common currency* | 721,149 | 0.0158 | 0.125 | 0 | 1 |

control for the economic scale effect. The economic scale effect (income-effect) of exporter/ importer countries (agreement partner) on bilateral trade flow is significantly positive. Surely economic scale effect raises the bilateral trade between partners. It is remarkable to note that the magnitude impact (0.31 for importers, 0.35 for exporters) of both partners is significant at the 1% level but slightly different. This implies that trade will increase at a significant level on both sides which ultimately increases the welfare gains for both countries. The positive impact of income was validated with Yao et al. [76]. However, geographical distance negatively influences bilateral trade flow, consistent with the traditional gravity model results. This implies that distance causes a decrease in bilateral trade due to the shipping cost of transferring goods from one country to another. Therefore, adjacent countries promote bilateral trade rather than distant countries. Consistently, our results confirmed that the adjacent countries contribute positively to increasing bilateral trade, as the coefficient (0.3975) impact is significantly positive. Communication is important for promoting trade between two countries [77]. For example, each country has its rules and regulations, which are settled between the two countries through communication. Therefore, a common language can be a powerful tool to deal with

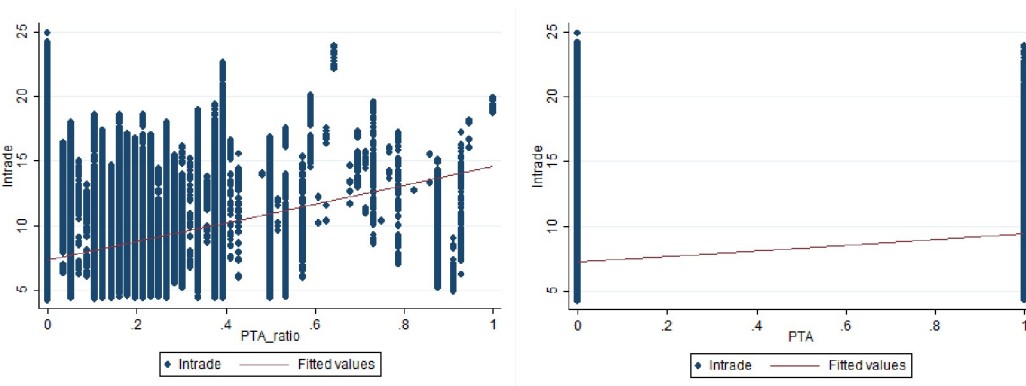

**Fig 4. Basic relationship between PTAs and trade.**

**Table 2. Empirical results of how PTAs affects bilateral trade.**

| Variables | MD | MD | MD | MD | MD | MD |
|---|---|---|---|---|---|---|
| | (1) | (2) | (3) | (4) | (5) | (6) |
| PTA | 0.1527*** | 0.0375*** | 0.0350*** | | | |
| | (0.024) | (0.007) | (0.005) | | | |
| PTA_Ratio | | | | 0.5711*** | 0.0320** | 0.0413*** |
| | | | | (0.071) | (0.014) | (0.010) |
| Lnyi | 0.3111*** | 0.2946*** | | 0.3133*** | 0.2941*** | |
| | (0.004) | (0.004) | | (0.004) | (0.004) | |
| Lnyj | 0.3585*** | 0.3498*** | | 0.3607*** | 0.3493*** | |
| | (0.005) | (0.005) | | (0.005) | (0.005) | |
| LnDis | -0.9903*** | | | -0.9876*** | | |
| | (0.017) | | | (0.017) | | |
| Contig | 0.3975*** | | | 0.4067*** | | |
| | (0.091) | | | (0.091) | | |
| Comlang | 0.1230*** | | | 0.1195*** | | |
| | (0.028) | | | (0.028) | | |
| Colony | 1.0761*** | | | 1.0821*** | | |
| | (0.093) | | | (0.093) | | |
| Comcur | 2.5615*** | 0.0782*** | 0.0346*** | 2.5605*** | 0.0764*** | 0.0313*** |
| | (0.147) | (0.010) | (0.009) | (0.147) | (0.010) | (0.009) |
| Year FE | Y | Y | | Y | Y | |
| Country FE | Y | | | Y | | |
| Country-pair FE | | Y | Y | | Y | Y |
| Country-year FE | | | Y | | | Y |
| Obs | 721,149 | 721,135 | 721,135 | 721,149 | 721,135 | 721,135 |
| $R^2$ | 0.8273 | 0.9896 | 0.9965 | 0.8274 | 0.9896 | 0.9965 |

Note:

*, **, and *** represent significance at the 10%, 5%, and 1% levels, respectively. The t-statistic of the robust standard deviation of the estimated coefficients is given in parentheses. The standard deviation is clustered at the country-pair level. MD = model.

these issues, that is, more communication in the same language, more understating more international supply chain and thereby more trade [76]. The colonial relationship also affects trade between two countries [77]. Our results also show the positive influence of colonial on trade follow. This means that the colonies strengthen the trade relationship. Currency helps to determine a country's economic health. The same currency would help promote trade between countries as trade becomes cheaper if trading partners have the same currency [78]. In addition, our results showed a positive impact on trade flow.

To control for factors that may disturb the estimation, we added year-fixed effects and country-fixed effects. In column (2) of Table 2, we added country-pair fixed effects to control for all potential factors between countries. In this regression, the impact of PTAs on bilateral trade is again positive and significant at 1% level. However, we found that the marginal effect of PTA on bilateral trade dropped from 0.165 (as shown in column 1) to 0.038 (as shown in column 2), which means that when we control for more paired country factors, the marginal effect decreases. The marginal effect is calculated by the equation: $(e^{coefficient} - 1)*100\%$. In addition, we found no change in the influence of exporter/importer countries income on bilateral trade flow as the coefficient impact is positive.

We further added the country-year fixed effects to control for the country-year level factors, which may disturb the estimation. The result in column (3), shows that the marginal effect of PTA on bilateral trade decreases further to 0.0356 while it is still significant at the 1% level. The marginal effect given by the structural gravity model indicates that the bilateral trade would increase by 3.56% if countries have RTA between them. Additionally, we investigated the impact of PTAs with the same currency holding other parameters constant. Results showed that PTAs increase bilateral trade flow if these countries have the same currency. Furthermore, a common currency is good for decreasing exchange rate volatility to zero [79, 80].

For the first three columns, we used the PTA dummies listed in the WTO. However, Baier et al. stated that dummy variables hardly capture the heterogeneous effect of PTAs [70]; hence, the marginal effect was not accurate. Therefore, we constructed the coverage index of PTAs. The results of the coverage PTAs index reported in column (4)–(6) of Table 2. The results showed that the impact of the coverage index is positive and significant at the 1% level. This implies that under the global production network background, the product-based coverage of PTAs can promote bilateral trade significantly. Similarly, others parameters promote bilateral trade. We tested the coverage index of PTAs impact on bilateral trade with economic scale holding others parameters constant. The results again validate the promoting effect of the coverage index on bilateral trade. The findings in the column (4) indicated that the coverage index of PTAs has a positive impact on bilateral trade. The same currency promote trade between the member countries. In conclusion, the baseline results of the structural gravity model showed that the bilateral trade would increase if two countries have a common trade agreement and the trade promotion effect is positively affected by the product-based coverage of PTAs suggesting that market access openness of PTAs can significantly promote bilateral trade, supporting the trade creation effect of trade agreement.

## 4.2 Robustness checks

Scholars have constructed other variables to measure the heterogeneity of PTAs according to the content of PTAs [22–24]. To distinguish our product-based coverage index of PTAs from their measures, we added these variables in the robustness checks. We first compared the product-based coverage index of PTAs with the depth and flexibility measures, and the results are shown in Table 3. We can conclude that the product-based coverage index of PTAs is positively correlated with the depth and flexibility of PTAs, with correlation coefficients of 0.327 and 0.65, respectively. The results show that there are differences between the product-based coverage index and the depth index (depth_rasch) in measuring the coverage of PTAs. The product-based coverage index mainly focuses on the products mentioned in the PTAs, while the depth index covers the seven key provisions. The correlation between the product-based coverage index and flexibility of PTAs (flexescape) is 0.65, suggesting that PTAs are more flexible if they cover more products. We checked the variance inflation factor (VIF) to make sure

**Table 3. Correlation matrix of different measures of PTAs.**

|  | PTA_ratio | depth_rasch | flexescape | VIF |
|---|---|---|---|---|
| PTA_ratio | 1.000 |  |  | 2.35 |
| depth_rasch | 0.327*** | 1.000 |  | 2.12 |
| flexescape | 0.650*** | -0.093*** | 1.000 | 1.37 |

Note: We calculated the correlation matrix using Pearson's method.

*, **, and *** represent 10%, 5%, and 1% significance levels, respectively. We use the bilateral trade variable as the independent variable and three measures of PTAs as the dependent variable to calculate the VIF value, which is shown in the last column.

**Table 4. Robustness checks of the product-based coverage index (PTA_ratio) on bilateral trade.**

| Variables | Full Sample | High-High | High-No High | No High-No High |
|---|---|---|---|---|
| | (1) | (2) | (3) | (4) |
| PTA_ratio | 0.0450*** | 0.0173 | 0.0163 | 0.0849*** |
| | (0.009) | (0.016) | (0.021) | (0.015) |
| depth_rasch | 0.0316*** | 0.0375*** | 0.0341*** | 0.0199*** |
| | (0.004) | (0.005) | (0.010) | (0.006) |
| flexescape | -0.0031* | -0.0098*** | -0.0070 | -0.0009 |
| | (0.002) | (0.003) | (0.004) | (0.003) |
| Country-pair FE | Y | Y | Y | Y |
| Country-year FE | Y | Y | Y | Y |
| Obs | 721,135 | 80,252 | 166,971 | 474,912 |
| $R^2$ | 0.9965 | 0.9985 | 0.9980 | 0.9959 |

Note: we divide our full sample into different samples based on whether the country is a high-income group. High- High denotes that both countries are of high income, High- No high means one country is of high income and the other is of middle or low income. No high -No high means that both countries are from the middle or income group.

that there is no multicollinearity problem. The results in the last column of Table 3 shows that there is no multicollinearity problem with the VIF values of the three variables below 10.

We then added these measures of PTAs in our robustness checks to test the effect of the product-based coverage index of PTAs on bilateral trade. The empirical results in column (1) of Table 4 show that the coverage index of PTAs has a significant positive effect on bilateral trade. Column (1) shows the robustness check of the baseline models, with the depth and flexibility measures of PTAs controlled. We also find that the depth of PTAs can promote bilateral trade, but the marginal effect (0.0316) is less than the coverage index (0.0450). The flexibility of PTAs negatively affects bilateral trade, because the PTAs are flexible to adjust their policies for other purposes without violating the terms of an agreement. They could not cut the tariff as promised because they could withdraw the PTAs. This is important for countries to design flexible PTAs. Flexible PTAs provide for legally accepted opt-outs without leading to a de jure breach of an agreement and encompass exit options, duration and renegotiation clauses, reservations, escape clauses, and withdrawal clauses, and hence help members to cooperate better [23]. The product-based coverage of PTAs differs them in focusing on the market access openness of PTAs instead of other aspects.

## 4.3 Heterogeneity analysis of different countries

We further analyzed heterogeneous effect of the product-based coverage index of PTAs on bilateral trade. We used the World Development Indicators (WDI, available at: https://datacatalog.worldbank.org/dataset/world-development-indicators) to identify whether a country is high-income and then we divided the full sample into different groups. The results are presented in column (2)–(4) of Table 4. Column (2) and (3) show that the coverage index of PTAs is insignificant if the country is from a high-income group. However, the depth of the PTAs is still significant at the 1% level. When both countries are not high-income countries, the coverage index of PTAs is significant at the 1% level. Based on these results, we concluded that the product-based coverage index of PTAs could promote bilateral trade for both middle-income and low-income groups, because liberalizing market access barriers is crucial for these countries. However, for high-income countries, market access barriers are already low, and the depth of PTAs is more important to promote bilateral trade. The results shed light on the

globalization development for different countries. High-income countries should focus on other aspects of PTAs which can contribute to higher quality trade. However, the less developed countries should liberalize the product market gradually to promote bilateral trade.

## 4.4 Effect of product-based coverage index on trade components

With the increasing development of GVCs, our measure based on the product covered in PTAs is important for studying the effect of PTAs on bilateral trade. Further, we use the EORA input-output database to calculate bilateral trade volume and decompose total trade into components in terms of types of trade (intermediate goods, final goods, commodities, and service trade). We investigate the product-based coverage index of PTAs on these different trade types. The empirical results are presented in Table 5. The findings of column (1) and (2) show that PTAs with a higher degree of openness have a significant positive promotional impact on both intermediate and final goods trade. As the estimated coefficients are 0.0408 and 0.030, which means that when the product-based coverage of the PTAs is 1, the maximum promotion effect of the implementation of the PTAs on intermediate goods trade is 4.185%. The maximum promotion effect on final product trade is 3.045%. PTAs can help countries sell final products and hence promote the welfare of countries, and be embedded in the GVCs through intermediate goods trade, suggesting that the product liberalization brought by PTAs can promote the development of the global production network. Countries can sign trade agreements covering a wider range of products to better integrate into the GVCs and promote their trade development levels. Further, the findings of the differential impact of PTA_ratio on service trade and final goods trade are reported in columns (3) and (4) of Table 5. The results showed that regional service trade agreements significantly promote trade flow, with coefficients of 0.0436 and 0.0411, respectively. This implies that the maximum promotion effect of PTAs on services trade is 5.527%, and the maximum promotion effect on trade goods is 4.185%. PTAs cover tariff barriers and the 'new age' matters such as foreign direct investment, services, labor and environmental standards. The service trade promotion of PTAs shows that high-standard PTAs help countries open up more broadly and improve the overall trade status.

**Table 5. Empirical results of the coverage index (PTA_ratio) on trade components.**

| Variables | lnmid_trade | lnfd_trade | lnservice | lncommo |
|---|---|---|---|---|
| | **(1)** | **(2)** | **(3)** | **(4)** |
| PTA_ratio | 0.0387*** | 0.0509*** | 0.0538*** | 0.0334*** |
| | (0.009) | (0.010) | (0.011) | (0.008) |
| depth_rasch | 0.0313*** | 0.0263*** | 0.0420*** | 0.0153*** |
| | (0.003) | (0.004) | (0.004) | (0.003) |
| flexescape | -0.0020 | -0.0043** | -0.0062*** | 0.0025* |
| | (0.002) | (0.002) | (0.002) | (0.001) |
| Comcur | 0.0210** | 0.0282** | 0.0232** | -0.0036 |
| | (0.009) | (0.011) | (0.012) | (0.007) |
| Country-pair FE | Y | Y | Y | Y |
| Country-year FE | Y | Y | Y | Y |
| Obs | 721,135 | 721,135 | 721,135 | 721,135 |
| $R^2$ | 0.9966 | 0.9961 | 0.9955 | 0.9975 |

Note:

*, **, and *** represent significance at the 10%, 5%, and 1% levels, respectively. The t-statistic of the robust standard deviation of the estimated coefficients is given in parentheses. The standard deviation is clustered at the country-pair level. The independent variables of columns (1)–(4) are intermediate goods trade, final foods trade, services trade, and commodity trade. Country-pair and country-year fixed effects are included in all regressions.

## Conclusion

Previous studies have generally proved the effect of PTAs on trade promotion between signatories. However, a few studies have considered the heterogeneity of agreement, especially to distinguish the market access levels of different PTAs. In contrast, this study identifies the product-based coverage of PTAs and investigates the effect of the product-based coverage index of PTAs on bilateral trade. This study compares the effect of the product-based coverage of PTAs with other depth measures and finds that the product-based coverage promotes bilateral trade for less developed countries, while the depth measure matters for developed countries. Finally, we decompose the total trade volume into different types of trade to examine the effect of the product-based coverage index of PTAs on GVCs and find that the product liberalization brought by PTAs can promote the development of the global production network.

The main conclusions of the study are as follows: (1) From the perspective of the market access level of PTAs, the product-based coverage index of PTAs between countries significantly promotes the import and export trade between directly contributing countries (including total trade volume, intermediate and final goods trade, service trade, and goods trade). (2) Compared with the depth of PTAs, the product-based coverage index of PTAs can only promote bilateral trade for countries from middle-income and low-income groups. (3) From the GVCs perspective, a wider range of product liberalization brought by PTAs can promote the development of global production network.

The study has some relevant policy implications regarding PTAs and bilateral trade: The product-based coverage index of PTAs between countries has a direct role in promoting bilateral trade. Countries should actively integrate into trade cooperation by signing high-standard PTAs. Additionally, when partners sign the PTAs, countries can consider two factors: (1) they choose from their existing friends that may share the short political distance with them; (2) they consider the friends of their existing PTA partners. In this way, they can turn the indirect relationship brought by PTAs into a direct relationship and avoid the externalities that the PTAs may bring.

For developing countries, the design of PTAs should continue to focus on gradually boosting the market access level. The participation of developing countries in the global PTA network should further enhance. Developing countries should expand the PTA network from two aspects: First, neighboring countries already have good relations. Second, countries with indirect ties with countries in the existing trade agreement network. Third, developing countries should strengthen the legal system to increase the operational efficiency of the judicial system for the smooth implementation and liberalization of PTAs.

For developed countries, the market access is already of high level, and promoting the coverage index of traditional trade cannot significantly boost bilateral trade. The design of PTAs for them should focus on the cooperation beyond tariff reductions, in services trade, investments, standards, public procurement, competition and intellectual property rights. Moreover, when signing the high- depth PTAs, they should consider that although the flexible PTAs can benefit the members in multiple ways, it could also affect bilateral trade because the countries can withdraw legally. Therefore, they should find a way to balance the side effects of flexible PTAs.

## Supporting information

**S1 Data.**
(DTA)

## Acknowledgments

We thank to the anonymous reviewers for their careful reading of our manuscript and their insightful comments and suggestions.

## Author Contributions

**Conceptualization:** Xing Yao, Yongzhong Zhang, Rizwana Yasmeen.

**Data curation:** Yongzhong Zhang, Zhen Cai.

**Formal analysis:** Yongzhong Zhang.

**Funding acquisition:** Xing Yao.

**Investigation:** Yongzhong Zhang, Zhen Cai.

**Resources:** Zhen Cai.

**Software:** Yongzhong Zhang.

**Supervision:** Xing Yao.

**Validation:** Rizwana Yasmeen, Zhen Cai.

**Visualization:** Yongzhong Zhang, Rizwana Yasmeen.

**Writing – original draft:** Yongzhong Zhang, Rizwana Yasmeen.

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
