## [Decision Letter · Decision Letter 0]

16 Sep 2020

PONE-D-20-21416

The impact of Regional Trade Agreements on Bilateral Trade: A Structural Gravity Model Analysis

PLOS ONE

Dear Dr. Yasmeen,

Thank you for submitting your manuscript to PLOS ONE. After careful consideration, we feel that it has merit but does not meet PLOS ONE’s publication criteria as it currently stands. Therefore, we invite you to submit a revised version of the manuscript that addresses the points raised during the review process.

Based on my own reading and recommendations from two excellent reviewers, I must admit that this was a close call, given that the reviews raise several concerns about the manuscript. A key weakness of the paper is its lack of clarity around operationalization and motivation for key measures. I therefore expect you to revise your manuscript taking all reviewer suggestions into account. 

Some specific guidance:

* Your introduction must be more effective. It should already mention how you define your coverage indicator, as main innovation to the research on RTA effectiveness. 

* Clearly identify the gap from existing research. Other papers have shown that RTAs can promote bilateral trade. What do we not know and why should we read your paper? 

* Your language needs to be more precise. This is not merely a matter of "poor translation" but involves an effort from your side to better explain what you do and why you do so. You must assume that readers will not know awfully much about RTAs. The reviewers rightfully question some of your modeling choices and you should give better explanations for them.

We look forward to receiving your revised manuscript.

Kind regards,

Bernhard Reinsberg, Ph.D

Academic Editor

PLOS ONE

Journal Requirements:

2. Please include a copy of Table 5 which you refer to in your text on page 18

Reviewers' comments:

Reviewer's Responses to Questions

**Comments to the Author**

1. Is the manuscript technically sound, and do the data support the conclusions?

Reviewer #1: Partly

Reviewer #2: Partly

2. Has the statistical analysis been performed appropriately and rigorously? 

Reviewer #1: No

Reviewer #2: No

3. Have the authors made all data underlying the findings in their manuscript fully available?

Reviewer #1: Yes

Reviewer #2: Yes

4. Is the manuscript presented in an intelligible fashion and written in standard English?

Reviewer #1: No

Reviewer #2: No

5. Review Comments to the Author

Reviewer #1: Summary of the article

The objective of this research article is to explore the impact of regional trade agreements (RTAs) on countries’ bilateral trade volume. The authors employ a structural gravity model and analyse 189 countries over the time period from 1990 to 2015. The principle finding of this article is that RTAs increase countries’ bilateral trade volume.

General comments

This article has a relatively clear structure but, in the view of the reviewer, does not comply with the Criteria for Publication 4 and 5. The language is unclear and often ambiguous. Central concepts such as, for instance, the “coverage ratio index” are also referred to as “coverage trade index”, “coverage RTAs index”, “coverage index”, etc. Acronyms are often not defined and y-axes in the figures are not labelled.

The authors state that their article makes three important contributions. First, the empirical analysis is claimed to account for the heterogeneity of RTA design by developing a RTA coverage index based on the World Trade Organization (WTO) Regional Trade Agreements Database. Unfortunately, however, the authors do neither discuss the existing literature on the heterogeneity of RTA design (e.g. Dür et al., 2014; Baccini et al., 2015; Hofmann et al., 2019) nor the alternative data sources (e.g. Dür et al., 2014; Matto et al., 2020) on RTA design. The literature review would benefit from a more elaborate discussion of these and other previous contributions. Furthermore, it would be interesting to see how the empirical results compare between the authors’ proposed coverage ratio index and the existing indicators of RTA design (e.g. depth, flexibility, scope). At the very least, these indicators should be included as control variables in the empirical analysis. If the reviewer understands correctly, one of the main findings of the article is that RTAs which cover more industries, also generate larger increases in trade volumes. To the reviewer, this makes sense but is not a particularly intriguing finding.

According to the authors, the second contribution of this article is related to the use of the EORA input-output database and the division of trade data into intermediate trade, final goods, services trade and commodity trade. Using such fine-grained trade data is a commendable effort that deserves a more detailed discussion and interpretation throughout the article.

The authors see the third contribution of the article in the use of political distance as an instrumental variable. While the construction of this variable is discussed in the Results and Discussion section, it would be helpful for the reader to learn more about this variable in the Data and Variables section.

Specific comments

1. Figure 1 is missing y-axis labels. It is not obvious to the reviewer, what the first y-axis shows and what the second y-axis shows. Page 3.

2. The regional distribution of RTAs is discussed in the context of Figure 2. The authors may also consider including regional variables in their empirical analysis to control for regional patterns. Page 3.

3. It is not clear to the reviewer why China is singled out and discussed in the context of Figure 3. From the abstract and the introduction, it appears to the reviewer that this is a general article and not specific to China or any other particular country. If China is of particular interest to the authors, they may consider explaining their case selection in more detail and including a binary variable for China RTAs in the empirical analysis. Page 3 and 4 and 14.

4. The authors may consider defining the acronyms the first time they appear in the text. While most readers in the field of international trade will be familiar with the used acronyms, other scholars might not be familiar with SAFTA, AFTA and BIMSTEC etc. Page 4.

5. As previously mentioned, the literature review is limited to gravity model contributions but insufficiently discusses scholarly work on RTA design and RTAs as 'Stepping Stones' vs. 'Stumbling Blocks'. Page 4.

6. The authors propose a coverage ratio index (which, confusingly, they sometimes refer to as a coverage trade index, or similar versions thereof) which the reviewer understands to be equal to the share of industries covered by the RTA. If this is correct, then one of the main findings of this paper is that the more industries are covered in a RTA, the larger the effect of the RTA on bilateral trade flows. To the reviewer, this makes sense and is not necessarily surprising. The authors may consider explaining in more detail what the contribution of this finding is. Page 6.

7. Figure 4 is missing a y-axis label. Page 6.

8. The authors may consider providing one or two examples of the economic organizations discussed on Page 7.

9. Some of the descriptive statistics in Table 1 are surprising to the reviewer. The coverage ratio index average, for instance, is 0.0379. This is far below the minimum point shown in Figure 4 (which is at around 0.2). It is also not clear to the reviewer, why the geographical distance is provided as a percentage. Page 8.

10. Figure 6 is missing y-axis labels.

11. The construction of the instrumental variable should be provided earlier on in the article. It is unclear to the reviewer, why a time lag of six periods in used. Page 12.

References

Baccini, L., Dür, A., and Elsig, M. (2015). The Politics of Trade Agreement Design: Revisiting the Depth-Flexibility Nexus. International Studies Quarterly, 59(4):765–775.

Dür, A., Baccini, L., and Elsig, M. (2014). The Design of International Trade Agreements: Introducing a New Dataset. The Review of International Organizations, 9(3):353–375.

Hofmann, C., Osnago, A., and Ruta, M. (2019). The Content of Preferential Trade Agreements. World Trade Review, 18(3):365-398.

Mattoo, A., Rocha, N., and Ruta, M. (2020). The Handbook of Deep Trade Agreements. Wahsington, DC: World Bank.

Reviewer #2: This paper uses a standard gravity model to analyze the effects of regional trade agreements (RTAs) on changes to bilateral trade volumes. The authors use an interesting dataset about the UN General Assembly Voting, political distance to capture the impact of RTAs on bilateral trade. The authors find that countries sharing a common RTA actually increases their trade volumes compared to those without RTAs, and they claim that there are welfare gains from signing RTAs.

Given the difficulty of the research question, I believe that the authors need to be more rigorous on how they approach the problem and clearly write down what they are finding. Most of my comments below are on the methodology and the data, and the structure of the paper.

Comments

1. In general, it is difficult to understand the main contributions of the paper. The research question is trying to understand the impact of RTAs on bilateral trade volumes. However, it is unclear to me what is really missing in the literature, and what the main contributions to the literature are. For example, calculating various trade volumes, the authors claim, is one of the main contributions. However, I do not see a good link between constructing different trade volumes and the main research question of the paper.

2. It would be good if the authors present the data clearly. For example, it is unclear from reading the paper how the data on UN voting looks like and what kind of information it gives us. This seems to be a crucial IV that the authors test in the latter section, and it should be clearly defined and also be explained why this was chosen to be the IV instead of other potential variables.

3. The authors claim that “the coverage index is more accurate in terms of the marginal trade creation effect of the trade agreement.” (line 335-line 336) I was not able to find further support for why the coverage index would be more accurate by reading the paper. Backing this claim is important because this is directly related to the main findings of the authors’ paper.

Furthermore, it would be good to show the intuition behind why the effects of RTAs (regression coefficient) jump from a coefficient of 0.15 to a coefficient of 0.57 when the authors include the RTA coverage index. Then the authors immediately conclude that (line 323-325) the RTA coverage index shows more accurate strength of RTAs. I think that there is a missing gap here, as it is not convincing why the effects more than quadruple. In fact, when the authors add country-pair fixed effects (γ_ij), the effects for RTA dummies and RTA coverage ratios become quite similar.

4. Understanding the content of the paper was difficult because there were frequent mechanical errors in the sentence structures. In fact, the main messages that the authors are trying to deliver are diluted as the paper is hard to comprehend.

6. PLOS authors have the option to publish the peer review history of their article (what does this mean?). If published, this will include your full peer review and any attached files.

Reviewer #1: No

Reviewer #2: No

---

## [Author Response · Author response to Decision Letter 0]

29 Oct 2020

Reply to Reviewers Comments for Paper PONE-D-20-21416

“The impact of Regional Trade Agreements on Bilateral Trade: A Structural Gravity Model Analysis”

Dear Editor and Reviewers,

We would like to commence by thanking the editor and the reviewers for their valuable time and constructive comments. Their expert knowledge of the field has helped us to strengthen the manuscript significantly. According to the valuable suggestions provided by the reviewers, we have revised the manuscript. We endeavored to address all the comments and our reflections are given below point by point.

Sincerely,

The Authors

Response to Reviewer One’s Comments

Reviewer #1: 

Reviewers' general comments:

This article has a relatively clear structure but, in the view of the reviewer, does not comply with the Criteria for Publication 4 and 5. The language is unclear and often ambiguous. Central concepts such as, for instance, the “coverage ratio index” are also referred to as “coverage trade index”, “coverage RTAs index”, “coverage index”, etc. Acronyms are often not defined and y-axes in the figures are not labelled.

The authors state that their article makes three important contributions. First, the empirical analysis is claimed to account for the heterogeneity of RTA design by developing a RTA coverage index based on the World Trade Organization (WTO) Regional Trade Agreements Database. Unfortunately, however, the authors do neither discuss the existing literature on the heterogeneity of RTA design (e.g. Dür et al., 2014; Baccini et al., 2015; Hofmann et al., 2019) nor the alternative data sources (e.g. Dür et al., 2014; Matto et al., 2020) on RTA design. The literature review would benefit from a more elaborate discussion of these and other previous contributions. Furthermore, it would be interesting to see how the empirical results compare between the authors’ proposed coverage ratio index and the existing indicators of RTA design (e.g. depth, flexibility, scope). At the very least, these indicators should be included as control variables in the empirical analysis. If the reviewer understands correctly, one of the main findings of the article is that RTAs which cover more industries, also generate larger increases in trade volumes. To the reviewer, this makes sense but is not a particularly intriguing finding.

According to the authors, the second contribution of this article is related to the use of the EORA input-output database and the division of trade data into intermediate trade, final goods, services trade and commodity trade. Using such fine-grained trade data is a commendable effort that deserves a more detailed discussion and interpretation throughout the article.

The authors see the third contribution of the article in the use of political distance as an instrumental variable. While the construction of this variable is discussed in the Results and Discussion section, it would be helpful for the reader to learn more about this variable in the Data and Variables section.

Response: Thank you very much for your valuable suggestions. In this new version, we have significantly improve the language of the entire manuscript, and believe this new version is more readable. In this new version, we have fully adopted your suggestions, especially the literature on the heterogeneity of RTAs design, which helps us enrich the literature review of this paper, so as to more clearly recognize the positioning and potential contribution of this paper in the literature. As for the explanation and elaboration of the data in this paper, this paper also makes a further supplement, especially for the calculation of the coverage index of RTAs and the instrumental variable we use, we describe it in more detail in Data and Variables section. Furthermore, we put all of our data sources in this version. 

Specific comments:

1: Figure 1 is missing y-axis labels. It is not obvious to the reviewer, what the first y-axis shows and what the second y-axis shows. Page 3.

Response: Thank you for the valuable comments and suggestions. We give a more detailed description of Figure 1. 

Figure 1 Global trend of RTAs over the years of 1989 – 2019. The y-axis on the left shows the number of different types (good notifications, services notifications and accessions to an RTA) of RTAs per year, and the y-axis on the right shows the cumulative notifications of RTAs in force and the cumulative number of RTAs in force.

2: The regional distribution of RTAs is discussed in the context of Figure 2. The authors may also consider including regional variables in their empirical analysis to control for regional patterns. Page 3.

Response: Thank you for the valuable comments and suggestions. In this new version, we try to add the regional variables in our empirical model (Whether a country locates in Europe or Asia). However, we already added country-pair fixed effects to control all the time-invariant country-pair factors, and the regional time-invariant variables, hence they will be omitted. The heterogenous effect of RTAs on countries from different group is necessary, therefore we used WDI database to divide country into two groups based on whether the country is with “High income”. Then we divided our sample into full groups according to whether a country is from “High income” group. The empirical results are as follows:

We further analyzed the market access effect of RTAs that could be different in terms of the economic development. We used World Development Indicators (WDI) to identify whether a country is High-income. Then we divided the full sample into different groups. The results of column (2) and (3) showed that the coverage index of RTAs is insignificant if the country is from high-income group. However, the depth of RTAs is still significant at 1% level. When both countries are not high-income countries, the coverage index of RTAs is significant at 1% level. We can conclude that the coverage index of RTAs can promote bilateral trade both for middle-income or low-income groups, because, to liberalize market access barriers is crucial for these countries. However, for high-income countries, the market access barriers are already less and the depth of RTAs is much important to promote bilateral trade.

Table 3

 Robustness checks of the coverage index (RTA_ratio) on bilateral trade 

Variables Full Sample

(1) High-High

（2） High-No High（3） No High-No High

(4)

RTA_ratio 0.0450*** 0.0173 0.0163 0.0849***

 (0.009) (0.016) (0.021) (0.015)

depth_rasch 0.0316*** 0.0375*** 0.0341*** 0.0199***

 (0.004) (0.005) (0.010) (0.006)

flexescape -0.0031* -0.0098*** -0.0070 -0.0009

 (0.002) (0.003) (0.004) (0.003)

Y Y Y Y

Y Y Y Y

Obs 721,135 80,252 166,971 474,912

R2 0.9965 0.9985 0.9980 0.9959

Note: See note under Table 2

3: It is not clear to the reviewer why China is singled out and discussed in the context of Figure 3. From the abstract and the introduction, it appears to the reviewer that this is a general article and not specific to China or any other particular country. If China is of particular interest to the authors, they may consider explaining their case selection in more detail and including a binary variable for China RTAs in the empirical analysis. Page 3 and 4 and 14.

Response: Thank you for the valuable comments and suggestions. we deleted the Figure 

4: The authors may consider defining the acronyms the first time they appear in the text. While most readers in the field of international trade will be familiar with the used acronyms, other scholars might not be familiar with SAFTA, AFTA and BIMSTEC etc. Page 4.

Response: Thank you for the valuable comments and suggestions. In this new version, we take this advice and define it the acronyms the first time they appear in the text. 

It is obvious in the era of globalization that international trade is a significant way to achieve long-term sustainable development [1]. Simultaneously trade agreements are the building blocks to raise the trade-integration among the nations by removing the trade barriers [2]. Thus, developed and developing countries are liberalizing their trade system by signing bilateral and regional preferential trade agreements [3]. For example, South Asian Association for Regional Cooperation (SAARC) member has formed the Bay of Bengal Initiative for Multi-Sectoral Technical and Economic Cooperation (BIMSTEC) in 2004 to promote economic integration among the countries. In 2006, SAARC member formed their RTAs (SAFTA—South Asian Free Trade Area) and ASEAN (Association of Southeast Asian Nation) member countries established AFTA (ASEAN Free Trade Area) in 1992 to boost trade among the member countries. According to the WTO database, the numbers of regional trade agreements were 82 in 2000; while in 2019, the numbers of regional trade agreements were 302, nearly four times higher than in 2000. 

5: As previously mentioned, the literature review is limited to gravity model contributions but insufficiently discusses scholarly work on RTA design and RTAs as 'Stepping Stones' vs. 'Stumbling Blocks'. Page 4.

Response: Thank you for the valuable comments and suggestions. In this new version, we added the existing literature referring to the RTA design and RTAs as 'Stepping Stones' vs. 'Stumbling Blocks'. RTA design is important and we should introduce in the Introduction section and compare our variable with the existing variables, and hence conclude our contribution clearly The contents are as follows:

This study based on the existing studies referring to the design of RTAs and the heterogeneous effect of RTAs on bilateral trade. Many scholars have constructed other variables to measure the heterogeneity of RTAs according to the content of RTAs [22-24]. The purpose of RTAs not only to create market access between members but also to establish broader economic integration rights in goods, services, and factor markets [25]. Dür et al. (2014) [22] used two different measures to operationalize the depth of RTAs. The first measure of depth is an additive index that combines seven key provisions that can be included in RTAs, and the second one relies on latent trait analysis (see Dür et al. 2014 [22]). Baccini et al. (2015) [23] measured the flexibility of RTAs and find a positive relationship between depth and flexibility for preferential trade agreements (PTAs). Hofmann et al. (2019) [24] offered a detailed assessment of preferential arrangement. They examined the coverage and legal enforceability of provisions regulating a large set of policy areas. Miroudot et al. (2010) [26] assessed RTAs through an analysis of market access and national treatment commitments at the level of 155 sub-sectors. However, they only analyzed 56 RTAs. Although some scholars have constructed the variables to measure the heterogeneity of RTAs, however, few of them explored the effect of different measure of RTAs on bilateral trade. However, we follow Miroudot et al. (2010) [26] and constructed the coverage index of RTAs by extracting all the products that are mentioned in each RTA document and match them to industry classification. We mainly focus on the effect of market access of RTAs on bilateral trade by controlling other measurements of RTAs. If an RTA covers more products from different sectors, then it will have a better market access effect. Then we analyze the coverage index of RTAs impact on bilateral trade by using the structural gravity model.

Though growing literature has focused on RTAs and their possibly welfare effects on the multilateral trading system. However, some researchers believed that bilateral free trade agreements are stepping-stones for multilateral trade liberalization, while others pointed out that bilateral trade agreements are obstacles against free trade [50]. Both believers whether the trade agreements undermine or propel the process of globalization backing their positions with logically consistent arguments. In the perspective of the obstacle [51-55] argued that RTAs acts as undermine towards multilateral growth and unleash a protectionist spiral. Further stated that RTAs effects like the spaghetti bowl effect and reduced the incentive for countries to enter into multilateral trade agreements. Regionalism, which leads to welfare losses in both member and excluded countries [56]. RTAs countries a lot oppose multilateral liberalization as they profit from the exclusion of competitors from their markets [57]. On the other side, scholars debated that RTAs acts as stepping-stones for further globalization [58-60]. In a Ministerial Meeting of the WTO in Doha (2001), governments expressed their opinion that “regional trade agreements can play an important role in promoting the liberalization and expansion of trade” [61]. Moreover, regional agreements backing the weak governments to implement reforms and make them stable even despite of domestic opposition. Partners possibly learn the advantages of liberalization once they practice limited free trade. 

To sum up, both effects (RTAs stumbling blocks & stepping –stones) are empirically relevant, which can capture the complexity of the effects of regional agreements for the process of globalization. Moreover, RTAs can encourage peaceful relationships by increasing the opportunity cost of conflicts [62].

6: The authors propose a coverage ratio index (which, confusingly, they sometimes refer to as a coverage trade index, or similar versions thereof) which the reviewer understands to be equal to the share of industries covered by the RTA. If this is correct, then one of the main findings of this paper is that the more industries are covered in a RTA, the larger the effect of the RTA on bilateral trade flows. To the reviewer, this makes sense and is not necessarily surprising. The authors may consider explaining in more detail what the contribution of this finding is. Page 6.

Response: Thank you for the valuable comments and suggestions. In this new version, we described in more detail that how the coverage index of RTA is constructed and the reason why we use it to measure the level of market access creation of RTAs. Our method is different from the traditional depth or flexibility measurement of RTAs that try to cover the different aspects of RTAs. The effect of the coverage index of RTAs on bilateral trade is robust when we control existing variables. We further find the heterogenous effect when we consider the difference of country pair. 

First, we clarify the contribution of our paper in the introduction section as follows:

This paper contributes to the RTAs and bilateral trade literature in the following ways. In contrast to previous literature, we construct the coverage index of RTAs using WTO regional trade agreements datasets to measure the level of market access of RTAs. This would be our valuable contribution to the literature as most of the study used dummies of regional trade agreements. Although some scholars measure the depth and flexibility of RTAs, few of them explore the heterogeneous effect of RTAs using different measures. We constructed the coverage index of RTAs and added other measurements in the regression, so we can explore the effect of market access on bilateral trade by controlling other variables. After that, we decompose the total trade into components and investigate the effect of the coverage of RTAs on different trade components. Specifically, this paper uses the EORA input-output database to calculate the bilateral trade flow between the agreements countries. The distinction of using the EORA database is that not only considered the intermediate goods between the national industries, but also considers the trade volume of final consumption. We divide the trade into Intermediate Trade, Final Goods Trade, Service Trade and Commodity Trade Goods to evaluate the impact of RTAs on them. Finally, we use lagged political distance as an instrumental variable to solve the endogenous issue. 

Second, we put the existing variables in our regression and analysis the differences between the coverage index of RTAs and the existing variables as follows:

Many scholars have constructed other variables to measure the heterogeneity of RTAs according to the content of RTAs [22-24]. We added these variables in our model and the empirical results showd that the coverage index of RTAs has a significant positive effect on bilateral trade in full sample, which is shown in column (1) of table 3. We also find that the depth of RTAs promote bilateral trade as well, but the marginal effect (0.0316) is less than the coverage index (0.0450). Moreover, the flexibility of RTAs has negative effect on bilateral trade, and the reason behind this is that the RTAs are flexible to adjust their policies for other purposes without violating the terms of an agreement. They could not cut the tariff as they promise because they can withdraw the RTAs. This is important for countries to design flexible RTAs. Flexible RTAs provide for legally accepted opt-outs without leading to a de jure breach of an agreement and encompass exit options, duration and renegotiation clauses, reservations, escape clauses, and withdrawal clauses, and hence help members to cooperate better [23].

We further analyzed the market access effect of RTAs. We used World Development Indicators (WDI) to identify whether a country is High-income and then we divided the full sample into different groups. The result presented in column (2) - (4) of Table 3. The results of column (2) and (3) showed that the coverage index of RTAs is insignificant if the country is from high-income group. However, the depth of RTAs is still significant at 1% level. When both countries are not high-income countries, the coverage index of RTAs is significant at 1% level. Based on results we concluded that the coverage index of RTAs could promote bilateral trade both for middle-income or low-income groups, because, to liberalize market access barriers is crucial for these countries. However, for the high-income countries, the market access barriers are already less and the depth of RTAs is much more important to promote bilateral trade.

Table 3

 Robustness checks of the coverage index (RTA_ratio) on bilateral trade 

Variables Full Sample

(1) High-High（2） High-No High（3） No High-No High

(4)

RTA_ratio 0.0450*** 0.0173 0.0163 0.0849***

 (0.009) (0.016) (0.021) (0.015)

depth_rasch 0.0316*** 0.0375*** 0.0341*** 0.0199***

 (0.004) (0.005) (0.010) (0.006)

flexescape -0.0031* -0.0098*** -0.0070 -0.0009

 (0.002) (0.003) (0.004) (0.003)

Y Y Y Y

Y Y Y Y

Obs 721,135 80,252 166,971 474,912

R2 0.9965 0.9985 0.9980 0.9959

Note: we divide our full sample into different samples based on whether the country is from high income group. High High denotes both countries are of high income，High Non high means one country is of high income and the other is of middle or low income. No high No high means both countries are from middle or income group

7: Figure 4 is missing a y-axis label. Page 6.

Response: Thank you for the valuable comments and suggestions. In this new version, we added the y-axis into previous Figure 4 and introduce the calculation in more detail. In this version, this Figure is labeled as Figure 3. 

Figure 3 The trend of coverage ratio index of RTAs over the years 2000 to 2019. The documents of RTAs are from WTO website: http://rtais.wto.org/UI/PublicMaintainRTAHome.aspx. We calculate the mean value of coverage index of RTAs in each year and we can see that the tread is upward.

8: The authors may consider providing one or two examples of the economic organizations discussed on Page 7.

Response: Thank you for the valuable comments and suggestions. In this new version, we fully follow your advice and take two examples to describe how we deal with the coverage index of RTA between two countries.

In multilateral agreements, a country and an economic organization divided into a country and the economic organization that shares the same RTA, for example, ASEAN. For any two countries in the ASEAN, they share the same RTAs and the coverage index of RTA. In 2002, CAFTA (China and ASEAN Free Trade Area) signed by China and ASEAN countries. In this case, China and each country of ASEAN share the same RTA and the coverage index of RTA. To conclude, if an agreement is sign within an economic organization, all countries included in the economic organization would be match with each other, and finally multiple countries will share the agreement. Similarly, we treat other multilateral agreements and get the reciprocal industry penetration of agreements countries. Based on the coverage index of RTAs, this paper studies the relationship between RTAs and bilateral trade.

9: Some of the descriptive statistics in Table 1 are surprising to the reviewer. The coverage ratio index average, for instance, is 0.0379. This is far below the minimum point shown in Figure 4 (which is at around 0.2). It is also not clear to the reviewer, why the geographical distance is provided as a percentage. Page 8.

Response: Thank you for the valuable comments and suggestions. In this new version, we recheck our data and method and rewrite the most of the paper according to reviewers’ suggestions. The coverage index average is 0.0379 as shown in Table1, this mean value is calculated using the whole observations, because many country-pair observations don’t share a common RTA, which means that the RTA_ratio=0 for those observations. However, the trend of RTA_ratio that are shown in Figure 4 is calculated using all RTAs in force. That’s why the mean value of RTA_ratio in Table 1 is not close with Figure 4. As for the geographical distance, we take a log of the actual geographical distance, so the mean value is 8.704 (the percentage is not proper here, thank you for your kind advice). 

The maximum and minimum income scale are 30.49 and 18.63 respectively. The Political distance average between trading countries is 0.793. While the mean value of the geographical distance is about 8.704. The relationship between trade and RTAs shown in Figure 5. From the figure, it is clear that RTAs and the coverage trade index have an upward trend. However, bilateral trade increase comparatively more by coverage trade index. It implies that RTAs decrease the tariff and promote trade between partner countries.

Table 1.

Description 

variables definition N Mean Sd Min Max

lntrade Total Trade value 721,149 7.699 2.892 4.268 24.86

lnmid_trade Intermediate goods trade value 721,149 7.350 2.795 3.930 24.86

lnfd_trade final goods trade value 721,149 6.250 3.153 2.488 23.55

lnservice service trade value 721,149 7.083 2.984 3.574 24.72

lncommo goods trade value 721,149 6.846 2.764 3.296 23.59

RTA_ ratio coverage index of RTAs 721,149 0.0379 0.111 0 1

RTA RTA Dummy 721,149 0.155 0.362 0 1

depth_rasch Depth of RTAs 721,149 -0.027 0.463 -1.433 2.267

flexescape Flexibility of RTAs 721,149 0.500 1.276 0 4

absidealdiff Political distance 721,149 0.793 0.795 0 4.887

lnYi/lnYj Log GDP 721,149 23.81 2.224 18.63 30.49

lnDis Log Distance 721,149 8.704 0.809 0.632 9.892

Contig Whether contig 721,149 0.0189 0.136 0 1

Comlang_ Whether common-language 721,149 0.134 0.341 0 1

Colony Whether colony 721,149 0.0124 0.111 0 1

Comcur Whether common currency 721,149 0.0158 0.125 0 1

10: Figure 6 is missing y-axis labels.

Response: Thank you for the valuable comments and suggestions. In this new version, we added y-axis. Figure 6 (in this version, it is Figure 5).

 Figure 5 Basic relation between RTAs and Trade

11: The construction of the instrumental variable should be provided earlier on in the article. It is unclear to the reviewer, why a time lag of six periods in used. Page 12.

Response: Thank you for the valuable comments and suggestions. In this new version, we rewrite the Data and Variables section. Firstly, we describe our data and variables in more detail, especially the instrumental variable. We use a time lag of six periods. We also check a time lag of five periods and the result does not show significant difference. We revise the content as follows:

(4) We constructed an instrumental variable to deal with the potential endogenous issue. Baccini et al. (2015) [23] used democracy to measure politics and proved that the politics have significant effect on the RTAs. Bailey et al. (2017) [73] used Votes in the United Nations General Assembly (UNGA) and developed a state-of-the-art ideal point model to estimate dynamic national ideal points along a single dimension from 1946 to 2012. By Following Bailey et al. (2017) [73], we used the United Nations voting database to measure the political distance between countries . Specifically, the database uses the distance between "ideal points" to describe the different voting options of countries in the UN General Assembly resolutions. We used a time lag of six periods of political distance as the IV for two concerns: one is that the political distance is a long-term factor and the time lag of six periods of political distance could affect a country’s decision on signing RTAs. However, the RTAs can hardly influence the time lag of six periods of political distance. Secondly, the less availability of UNGA data after 2009. We then used the time lag of six periods of political distance to use more regression observations .

 

Response to Reviewer Two’s Comments

Comments: Reviewer #2:

Reviewers' general comments:

This paper uses a standard gravity model to analyze the effects of regional trade agreements (RTAs) on changes to bilateral trade volumes. The authors use an interesting dataset about the UN General Assembly Voting, political distance to capture the impact of RTAs on bilateral trade. The authors find that countries sharing a common RTA actually increases their trade volumes compared to those without RTAs, and they claim that there are welfare gains from signing RTAs.

Given the difficulty of the research question, I believe that the authors need to be more rigorous on how they approach the problem and clearly write down what they are finding. Most of my comments below are on the methodology and the data, and the structure of the paper.

Response: Thank you very much for your valuable advice. In the process of revising the paper, we have fully adopted your suggestions. As for the explanation and elaboration of the data in this paper, this paper also makes a further supplement, especially for the calculation of the coverage index of RTAs and the instrumental variable, we described it in more detail in Data and Variables section. Furthermore, we put all of our data sources in this version. The response to specific comments are as follows.

Specific comments:

1: In general, it is difficult to understand the main contributions of the paper. The research question is trying to understand the impact of RTAs on bilateral trade volumes. However, it is unclear to me what is really missing in the literature, and what the main contributions to the literature are. For example, calculating various trade volumes, the authors claim, is one of the main contributions. However, I do not see a good link between constructing different trade volumes and the main research question of the paper.

Response: Thank you for the valuable comments and suggestions. In this new version, we have significantly improve the writing of our paper, and we introduce our contribution in the Introduction section as follows:

This study based on the existing studies referring to the design of RTAs and the heterogeneous effect of RTAs on bilateral trade. Many scholars have constructed other variables to measure the heterogeneity of RTAs according to the content of RTAs [22-24]. The purpose of RTAs not only to create market access between members but also to establish broader economic integration rights in goods, services, and factor markets [25]. Dür et al. (2014) [22] used two different measures to operationalize the depth of RTAs. The first measure of depth is an additive index that combines seven key provisions that can be included in RTAs, and the second one relies on latent trait analysis (see Dür et al. 2014 [22]). Baccini et al. (2015) [23] measured the flexibility of RTAs and find a positive relationship between depth and flexibility for preferential trade agreements (PTAs). Hofmann et al. (2019) [24] offered a detailed assessment of preferential arrangement. They examined the coverage and legal enforceability of provisions regulating a large set of policy areas. Miroudot et al. (2010) [26] assessed RTAs through an analysis of market access and national treatment commitments at the level of 155 sub-sectors. However, they only analyzed 56 RTAs. Although some scholars have constructed the variables to measure the heterogeneity of RTAs, however, few of them explored the effect of different measure of RTAs on bilateral trade. However, we follow Miroudot et al. (2010) [26] and constrfucted the coverage index of RTAs by extracting all the products that are mentioned in each RTA document and match them to industry classification. We mainly focus on the effect of market access of RTAs on bilateral trade by controlling other measurements of RTAs. If an RTA covers more products from different sectors, then it will have a better market access effect. Then we analyze the coverage index of RTAs impact on bilateral trade by using the structural gravity model. 

Although, we applied the structural gravity model to deal with the bias estimation. However, the endogenous issue can still occur for the sake that countries choose their RTAs partners to get more benefits. Therefore, we adopt the lagged political distance as an IV which is measured by the absolute distance between country i and country j. Subsequent ideal point estimates are based on United Nations General Assembly Voting Data [27]. For that, the 2SLS estimation is strongly robust. Our result is stable and robust after considering the endogenous issues. 

 This paper contributes to the RTAs and bilateral trade literature in the following ways. In contrast to previous literature, we construct the coverage index of RTAs using WTO regional trade agreements datasets to measure the level of market access of RTAs. This would be our valuable contribution to the literature as most of the study used dummies of regional trade agreements. Although some scholars measure the depth and flexibility of RTAs, few of them explore the heterogeneous effect of RTAs using different measures. We constructed the coverage index of RTAs and added other measurements in the regression, so we can explore the effect of market access on bilateral trade by controlling other variables. After that, we decompose the total trade into components and investigate the effect of the coverage of RTAs on different trade components. Specifically, this paper uses the EORA input-output database to calculate the bilateral trade flow between the agreements countries. The distinction of using the EORA database is that not only considered the intermediate goods between the national industries, but also considers the trade volume of final consumption. We divide the trade into Intermediate Trade, Final Goods Trade, Service Trade and Commodity Trade Goods to evaluate the impact of RTAs on them. Finally, we use lagged political distance as an instrumental variable to solve the endogenous issue.

2: It would be good if the authors present the data clearly. For example, it is unclear from reading the paper how the data on UN voting looks like and what kind of information it gives us. This seems to be a crucial IV that the authors test in the latter section, and it should be clearly defined and also be explained why this was chosen to be the IV instead of other potential variables.

Response: Thank you for the valuable comments and suggestions. In this new version, we rewrite the Data and Variables section. Firstly, we described our data and variables in more detail, especially the instrumental variable. we use a time lag of six periods. We also check a time lag of five periods and the result does not show significant difference. We revise the content as follows:

(4) We constructed an instrumental variable to deal with the potential endogenous issue. Baccini et al. (2015) [23] used democracy to measure politics and proved that the politics have significant effect on the RTAs. Bailey et al. (2017) [73] used Votes in the United Nations General Assembly (UNGA) and developed a state-of-the-art ideal point model to estimate dynamic national ideal points along a single dimension from 1946 to 2012. By Following Bailey et al. (2017) [73], we used the United Nations voting database to measure the political distance between countries . Specifically, the database uses the distance between "ideal points" to describe the different voting options of countries in the UN General Assembly resolutions. We used a time lag of six periods of political distance as the IV for two concerns: one is that the political distance is a long-term factor and the time lag of six periods of political distance could affect a country’s decision on signing RTAs. However, the RTAs can hardly influence the time lag of six periods of political distance. Secondly, the less availability of UNGA data after 2009. We then used the time lag of six periods of political distance to use more regression observations .

3: The authors claim that “the coverage index is more accurate in terms of the marginal trade creation effect of the trade agreement.” (line 335-line 336) I was not able to find further support for why the coverage index would be more accurate by reading the paper. Backing this claim is important because this is directly related to the main findings of the authors’ paper.

Furthermore, it would be good to show the intuition behind why the effects of RTAs (regression coefficient) jump from a coefficient of 0.15 to a coefficient of 0.57 when the authors include the RTA coverage index. Then the authors immediately conclude that (line 323-325) the RTA coverage index shows more accurate strength of RTAs. I think that there is a missing gap here, as it is not convincing why the effects more than quadruple. In fact, when the authors add country-pair fixed effects (γ_ij), the effects for RTA dummies and RTA coverage ratios become quite similar.

Response: Thank you for the valuable comments and suggestions. In this new version, we described the detail how the coverage index of RTA is constructed and the reason why we use it to measure the level of market access creation of RTAs. Our method is different from the traditional depth or flexibility measurement of RTAs, which try to cover the different aspects of RTAs. And the effect of the coverage index of RTAs on bilateral trade is robust when we control existing variables, and we further find heterogenous effect when we consider the difference of country pair. 

Specifically, we revised our explanation of the difference between RTA dummy and the coverage index of RTAs. The dummy can only capture the average effect of RTA on bilateral trade, but cannot measure the heterogeneous effect of different RTAs, Therefore, our measurement is good. Furthermore, scholars have constructed other variables to measure the heterogeneity of RTAs, we differ from them in indicating the market access level instead of other aspects, the content is as follows:

For the first three columns, we used RTAs dummies listed in WTO. However, Baier et al. (2019) stated that dummy variables hardly capture the heterogeneous effect of RTAs (see Baier et al. 2019) [70], and hence the marginal effect is not accurate. Therefore, we constructed the coverage index of RTAs as we have mentioned above. The results of the coverage RTAs index reported in column (4)-(6) of Table 5. The results showed that the impact of the coverage index is positive and significant at 1% level. Though the direction impact is not different but the magnitude, impact is higher than the RTAs dummies (as shown in column (3) and (6)). It implies that using the coverage index to measure the market access level of RTAs, the marginal effect is larger than the average effect of the RTA dummy, and hence we can have a better understanding of the effect of RTAs to increase the bilateral trade. In the presence of coverage trade index again we find that more income more bilateral trade. However, geographical reduce bilateral trade agreements. Similarly above, others parameters promote bilateral trade. We tested the coverage index of RTAs impact on bilateral trade with economic scale holding others parameters constant. Results again validate promoting effect of coverage index on bilateral trade. The findings of the column of (4) indicated that the coverage index of RTAs has a positive impact on bilateral trade. The same currency motivates for RTAs and thus trade increases between the signed countries. 

Scholars have constructed other variables to measure the heterogeneity of RTAs according to the content of RTAs [22-24]. We added these variables in our model and the empirical results showed that the coverage index of RTAs has significant positive effect on bilateral trade in full sample, which is shown in column (1) of table 3. We also find that the depth of RTAs can promote bilateral trade as well, but the marginal effect (0.0316) is less than the coverage index (0.0450). Moreover, the flexibility of RTAs has negative effect on bilateral trade, and the reason behind this is that the RTAs are flexible to adjust their policies for other purposes without violating the terms of an agreement. They could not cut the tariff as they promise because they can withdraw the RTAs. This is important for countries to design flexible RTAs. Flexible RTAs provide for legally accepted opt-outs without leading to a de jure breach of an agreement and encompass exit options, duration and renegotiation clauses, reservations, escape clauses, and withdrawal clauses, and hence help members to cooperate better [23].

We further analyzed the market access effect of RTAs. We used World Development Indicators (WDI) to identify whether a country is High-income and then we divided the full sample into different groups. The result presented in column (2) - (4) of Table 3. The results of column (2) and (3) showed that the coverage index of RTAs is insignificant if the country is from high-income group. However, the depth of RTAs is still significant at 1% level. When both countries are not high-income countries, the coverage index of RTAs is significant at 1% level. Based on results we concluded that the coverage index of RTAs could promote bilateral trade both for middle-income or low-income groups, because, to liberalize market access barriers is crucial for these countries. However, for the high-income countries, the market access barriers are already less and the depth of RTAs is much more important to promote bilateral trade.

Table 3

 Robustness checks of the coverage index (RTA_ratio) on bilateral trade 

Variables Full Sample

(1) High-High（2） High-NoHigh（3） NoHigh-NoHigh

(4)

RTA_ratio 0.0450*** 0.0173 0.0163 0.0849***

 (0.009) (0.016) (0.021) (0.015)

depth_rasch 0.0316*** 0.0375*** 0.0341*** 0.0199***

 (0.004) (0.005) (0.010) (0.006)

flexescape -0.0031* -0.0098*** -0.0070 -0.0009

 (0.002) (0.003) (0.004) (0.003)

Y Y Y Y

Y Y Y Y

Obs 721,135 80,252 166,971 474,912

R2 0.9965 0.9985 0.9980 0.9959

Note: we divide our full sample into different samples based on whether the country is from high income group. High -High denotes both countries are of high income，High- No high means one country is of high income and the other is of middle or low income. No high- No high means both countries are from middle or income group

In addition, we investigate the effect of RTAs on different trade types. We decompose the bilateral trade into intermediate goods trade, final goods trade, service trade and goods trade according to the difference in the final use of traded goods and the industry to which traded goods belong. The empirical results given in Table 3. The findings of column (1) and (2) showed that RTAs with a higher degree of openness have a significant positive promotional impact on both intermediate goods trade and final goods trade. As the estimated coefficients are 0.0408 and 0.030, which means that when the coverage of the RTAs is 1, the maximum promotion effect of the implementation of the RTAs on intermediate goods trade is 4.185%. The maximum promotion effect on the final product trade is 3.045%. This means that RTAs cannot only help countries to sell final products and hence promote the welfare of countries, but also embedded in the global value chain through intermediate goods trade.

4: Understanding the content of the paper was difficult because there were frequent mechanical errors in the sentence structures. In fact, the main messages that the authors are trying to deliver are diluted as the paper is hard to comprehend.

In this new version, we have significantly improve the language of the entire manuscript, and believe this new version is more readable.

---

## [Decision Letter · Decision Letter 1]

4 Dec 2020

PONE-D-20-21416R1

The impact of Regional Trade Agreements on Bilateral Trade: A Structural Gravity Model Analysis

PLOS ONE

Dear Dr. Yasmeen,

Thank you for submitting your manuscript to PLOS ONE. After careful consideration, we feel that it has merit but does not fully meet PLOS ONE’s publication criteria as it currently stands. Therefore, we invite you to submit a (major) revised version of the manuscript that addresses the points raised during the review process.

In all honesty, I was close to reject the paper based on comments received from the first-round reviewers. I have sent the manuscript out to a fresh reviewer, who have come back with excellent comments. Provided you could implement all their comments, this would bring the paper closer to publication but given the scope of the revisions needed, I cannot make a firm promise about this now. 

A *succinct* rebuttal letter that responds to each point raised by the academic editor and reviewer(s). You should upload this letter as a separate file labeled 'Response to Reviewers'.A marked-up copy of your manuscript that highlights changes made to the original version. You should upload this as a separate file labeled 'Revised Manuscript with Track Changes'.An unmarked version of your revised paper without tracked changes. You should upload this as a separate file labeled 'Manuscript'.

We look forward to receiving your revised manuscript.

Kind regards,

Bernhard Reinsberg, Ph.D

Academic Editor

PLOS ONE

Reviewers' comments:

Reviewer's Responses to Questions

**Comments to the Author**

1. If the authors have adequately addressed your comments raised in a previous round of review and you feel that this manuscript is now acceptable for publication, you may indicate that here to bypass the “Comments to the Author” section, enter your conflict of interest statement in the “Confidential to Editor” section, and submit your "Accept" recommendation.

Reviewer #1: (No Response)

Reviewer #3: (No Response)

2. Is the manuscript technically sound, and do the data support the conclusions?

Reviewer #1: No

Reviewer #3: Partly

3. Has the statistical analysis been performed appropriately and rigorously? 

Reviewer #1: I Don't Know

Reviewer #3: Yes

4. Have the authors made all data underlying the findings in their manuscript fully available?

Reviewer #1: Yes

Reviewer #3: Yes

5. Is the manuscript presented in an intelligible fashion and written in standard English?

Reviewer #1: No

Reviewer #3: Yes

6. Review Comments to the Author

Reviewer #1: The reviewer very much appreciates the authors' commendable efforts to provide detailed responses to the initial comments. However, unfortunately the revised text remains difficult to follow and the research question and scholarly contribution remain largely unclear. Furthermore, the presentation of the figures and tables leaves much to be desired.

Reviewer #3: Overall, I think this article has some potentially interesting contributions to make. That said, the authors need to do a better job communicating what those contributions are, illustrating the contribution empirically, and then situating them within the relevant literature(s). Moreover, the paper needs copyediting in order to make it clearer.

Major points:

On the whole I find the idea of constructing a product-based measure of the scope (or breadth) of PTAs and then relating this to trade flows to be a worthwhile endeavor. However, the paper needs to do a better job explaining: A) why one would want to do this, and B) what others have done in this regard. In general, I think the front end of the paper is not nearly crisp enough in this regard. You spend time talking about stumbling blocks versus stepping stones, which is not relevant at all, and much of the initial part of the paper goes over things we already know, such as the fact that PTAs have increased in number over time. I would instead have a more focused discussion at the front end of the paper that situates the current state of knowledge only with respect to PTAs/PTA design and trade flows. What have others done to assess the trade effects of PTAs and what are they potentially leaving out? Then you can offer your study as something that extends/enhances these previous studies and/or pushes it forward in unique ways.

For me one of the major contributions is the construction of your original variable RTA_ratio. As such, I think you need to motivate with this more and make very clear what this does that is different from previous studies. What does this measure tell us that other measures have not? How is different from or similar to the DESTA depth variables? I think you could make the argument that it similar conceptually to depth but is measured in a different way (which is a unique contribution). Or, alternatively, you could make the argument that it is different in important ways to depth (which is more about the specificity of commitments related to non-trade aspects of PTAs), and this helps us better understand the effect of PTAs on trade flows. But this needs to be clear.

Then, in the empirics, you then need to compare your variable to others, such as the depth variable. For example, are these two measures correlated and, if so, how? Depending on whether you view them as complementary or overlapping this will determine, for example, whether you should include them in the same empirical models. Right now, you have them in the same econometric models later in the paper, but if they are substitute measures, they should likely not be included for conceptual and empirical reasons (multicollinearity).

Another contribution seems to be that you are using a different dependent variable for some of your regressions, but you don’t really sell this that effectively. If this is novel, which I believe it is, then your paper is making an important contribution that RTAs with greater scope of products covered don’t just increase trade in general they especially increase trade in intermediate goods, which are important for supply chains. This is a good contribution, but it is hidden in the manuscript.

A final contribution is the finding that perhaps the trade creating effects of wider scope PTAs are most evident in developing country dyads. This is an important contribution too, but the writing here obscures the point. Make this clear. Also, try to make the language consistent with previous efforts.

Finally, I don’t really understand the rationale for the instrumental variable regressions. What type of endogeneity are you concerned with, and how is your selected IV going to address it? You mention the inclusion of democracy and other “political” variables in other papers, but these are often used as a control variable not in a 2 stage model. Moreover, the Baccini et al (2015) paper you cite is using joint democracy: A) as a control and B) in econometric models where the dependent variable is flexibility (from DESTA), not trade volumes. Overall, if you want to keep the IV regression, much more would need to be done to explain what value added it provides and then you need to show that the IV you select satisfies the exclusion restriction requirements.

Overall, I think the biggest issue(s) right now is with how you frame and set-up the research. The front end in particular – e.g. the intro and literature review sections – do not do a great job laying out what your paper is about and where it fits in.

Other points:

You never really discuss what the acronym EORA stands for or explain what this data is. I had to look it up.

First two sentences in section 2.1 imply that GATT/WTO and RTAs are equivalent somehow. PTAs are in fact antithetical to the core principles of GATT but are permissible under the rules if they satisfy certain conditions. More broadly the literature review should get away from basic stylized facts on PTAs (which should be common knowledge to people interested in the research) to talking more about what people have done conceptually and/or empirically to understand their effects on trade, which is the more disputed/evolving part of our understanding.

In figure 1, what is the difference between “cumulative notifications of RTAs in force” and “cumulative number of RTAs in force”? Do you mean signed versus ratified?

You adopt the term RTA throughout, but I think you are actually talking about preferential trade agreements (PTAs) not simply regional trade agreements (confusingly the WTO uses the RTA term, but I would still clarify this if you continue to use RTA). This also provides an entry to speak more to the fact that PTAs are very heterogenous, some cover a few products only (thinking of some middle eastern agreements), whereas some cover most or all of the trade between signatories.

I want to know more about the specifics of the construction of your variable RTA_ratio. You say that you constructed it from “information extract from the RTAs documents” on page 8, but you could go intro greater detail. If this is one of the central contributions then the steps you took need to be carefully explained and justified.

I will mention this again here. I think you need to compare DESTA’s depth variable and your RTA_ratio variable. Then, I would potentially run your econometric models’ side-by-side with one or the other, rather than models where both are included as in tables 3 and 4.

In table 2 I would write out what the different fixed effects terms rather than simply giving them their Greek symbols.

7. PLOS authors have the option to publish the peer review history of their article (what does this mean?). If published, this will include your full peer review and any attached files.

Reviewer #1: No

Reviewer #3: No

---

## [Author Response · Author response to Decision Letter 1]

17 Jan 2021

Reply to Reviewers Comments for Paper PONE-D-20-21416

“The impact of Regional Trade Agreements on Bilateral Trade: A Structural Gravity Model Analysis”

Dear Editor and Reviewers,

We would like to commence by thanking the editor and the reviewers for their valuable time and constructive comments. Their expert knowledge of the field has helped us to strengthen the manuscript significantly. We very much appreciate Reviewer#3’s valuable suggestions, which is really helpful for us to revise the manuscript. We endeavored to address all the comments and our reflections are given below point by point.

Sincerely,

The Authors

Response to Reviewer One’s Comments

Reviewer #1: 

Reviewers' general comments:

The reviewer very much appreciates the authors' commendable efforts to provide detailed responses to the initial comments. However, unfortunately the revised text remains difficult to follow and the research question and scholarly contribution remain largely unclear. Furthermore, the presentation of the figures and tables leaves much to be desired.

Response: Thank you very much for your valuable suggestions. In this new version, we have significantly improve the language of the entire manuscript, and believe this new version is more readable. In this revision, we rearrange our story and review the literature more carefully to clarify our contribution. Hopefully we can meet your demand and help you understand our contributions to existing literature.

Response to Reviewer #3’s Comments

Reviewer #3: 

Reviewers' general comments:

Overall, I think this article has some potentially interesting contributions to make. That said, the authors need to do a better job communicating what those contributions are, illustrating the contribution empirically, and then situating them within the relevant literature(s). Moreover, the paper needs copyediting in order to make it clearer.

Response: We very much appreciate your valuable suggestions, which we think are helpful for us to think more deeply about this study. We are happy that you acknowledge our contribution, and in this version, we really cherish your suggestions and revise the whole paper carefully, especially the introduction section, the literature section and the product-based coverage index of PTAs section. We also delete tables and figures which are not on the core of storyline. Hopefully we can clarify our contributions in this version. Again, thank you very much for your valuable suggestions. 

Specific comments:

1: On the whole I find the idea of constructing a product-based measure of the scope (or breadth) of PTAs and then relating this to trade flows to be a worthwhile endeavor. However, the paper needs to do a better job explaining: A) why one would want to do this, and B) what others have done in this regard. In general, I think the front end of the paper is not nearly crisp enough in this regard. You spend time talking about stumbling blocks versus stepping stones, which is not relevant at all, and much of the initial part of the paper goes over things we already know, such as the fact that PTAs have increased in number over time. I would instead have a more focused discussion at the front end of the paper that situates the current state of knowledge only with respect to PTAs/PTA design and trade flows. What have others done to assess the trade effects of PTAs and what are they potentially leaving out? Then you can offer your study as something that extends/enhances these previous studies and/or pushes it forward in unique ways.

Response: Thank you for the valuable comments and suggestions. We read our paper carefully and admit that the previous version does not link our contribution to existing literature well. In this version, we revise the introduction section and literature review section. And we focus on the design of PTAs and the effect of PTAs on bilateral trade. We further write in detail the difference of the product-based coverage index of PTAs and how we calculate this variable, to verify that the measure fits the reality, we take China as an example by comparing the object of the government while designing the PTAs and our measure of these PTAs, the results are reasonable. The following is the literature review section, from which hopefully we link our contribution with existing literature well.

2 Literature review 

In this section, we review related literature to clarify our contribution from the literature perspective. We mainly focus on two strands of literature: the effect of PTAs on bilateral trade and the design of PTAs.

Preferential trade agreements (PTAs) have been proliferating for the last twenty years. A large literature has studied various aspects of this phenomenon and study the design of PTAs. Some researchers early pointed out that bilateral trade agreements are obstacles against free trade [34]. In the perspective of the obstacle [35-39] argued that PTAs acts as undermine towards multilateral growth and unleash a protectionist spiral. They further stated that PTAs effects like the spaghetti bowl effect and reduced the incentive for countries to enter into multilateral trade agreements. Regionalism, which leads to welfare losses in both member and excluded countries [40]. In a Ministerial Meeting of the WTO in Doha (2001), governments expressed their opinion that “regional trade agreements can play an important role in promoting the liberalization and expansion of trade” [41]. The motivation of this evolution in PTAs is not only to ease tariff barriers but also the ‘new age’ matters such as foreign direct investment, services, labor and environmental standards [28]. More scholars applied the gravity model to estimate the effect of PTAs and find a positive effect on bilateral trade [39, 40]. However, most of the studies conceptualize PTAs as a dichotomous variable, namely whether countries sign an agreement or not, and hence treat PTAs as if they were all equal in purpose while estimating the effects of PTAs. The results are fruitful and they conclude that on average PTAs can promote bilateral trade for member countries. Some scholars estimated that the long-run effect of PTAs on bilateral trade flows is 100% and the effect varies considerably across trade agreements [42, 43]. Others applied the structural gravity model to analyze the PTAs and concluded that an average treatment effect of PTAs on trade flows is 236% [44]. Dembatapitiya and Weerahewa [45] investigated the SAFTA, European Union (EU), ASEAN, BIMSTEC and NAFTA regional trade impact and found mixed results. For example, the co-efficient for EU is significant, while, SAFTA, ASEAN, BIMSTEC and North American Free Trade Agreement (NAFTA) do not show a significant impact on bilateral trade. Sampson [46] study on the evolution of China’s PTAs and stated that China’s increasing network of PTAs is important and strategic for the Asia region. Pant and Paul [47] evaluated the PTAs for India and made an argument that PTAs are sounds to be good for the intra-regional trade volume and welfare of countries as well. Besides, scholars also assess the ex-post trade effects by applying the gravity model [48, 49]. Carre`re [50] studied ex-post PTAs and claimed that intra-regional trade mostly tied to a reduction in imports from the others countries of the world. However, Moreover, regional agreements backing the weak governments to implement reforms and make them stable even despite of domestic opposition. Partners possibly learn the advantages of liberalization once they practice limited free trade.

Until recently, scholars start to investigate the content and design of PTAs [51] and study regional specifications [52] or explain functional differences in design, for example with respect to dispute settlement [53] and flexibility provisions [54]. Furthermore, they found that not only tariff cuts, but also other market access and trade-related provisions in PTAs concerning topics such as investments and intellectual property rights matter for trade flows [5], and those deep agreements usually are flexible to adjust their policies for other purposes and withdraw the PTAs without violating the terms of an agreement [6]. Besides market access of goods, many PTAs today include provisions in such trade disciplines as services, investment, standards, intellectual property, and competition rules, as well as a host of issues not directly related to trade, such as the environment. At the beginning of the study of design of PTAs, Scholars usually focus on specific PTAs and measure the strength of a wide variety of provisions in the legal texts of PTAs [10,55]. With more and more documents of PTAs are available on WTO website, scholars extend their objects and started to pay greater attention to the scope and depth of these agreements [5-10,55-58]. They focus on the broader economic integration rights in goods, services, and factor markets brought by PTAs. For example, Dür et al. [5] used two different measures to operationalize the depth of PTAs. The first measure of depth is an additive index that combines seven key provisions that can be included in PTAs, and the second one relies on latent trait analysis. Baccini et al. [6] measured the flexibility of PTAs and find a positive relationship between depth and flexibility for preferential trade agreements (PTAs). Hofmann et al. [7] offered a detailed assessment of preferential arrangement. They examined the coverage and legal enforceability of provisions regulating a large set of policy areas. In addition to the depth of PTAs, many scholars study the breadth of PTAs [59-61]. Miroudot et al. [10] tried to construct industry-based coverage index of PTAs, however, they only study 56 service PTAs. Limão claim that a broader PTA is one where partners seek to increase market access not only in product markets, but also in markets for capital, labour, and technology [59]. Based on this, Hofmann et al. construct the measure based on a count of the provisions covered by a PTA and can focus either on the 18 ‘core’ provisions [60] and recent scholars also build a measure following this method [61]. Although some scholars have constructed the variables to measure the heterogeneity of PTAs, most of them focus on the non-trade aspects of PTAs and few of them measure the coverage index from the product-based perspective, which is important under the global production network background. 

In addition to the effect of PTAs on bilateral trade and the design of PTAs, this paper also investigates how PTAs affect intermediate good trade. Existing literature referring to global value chain (GVC) tend to study how deep trade agreements affect different type of trade recently [61-63]. Laget et al. [62] use trade data in both value-added and gross form to provide as complete a picture of GVC trade as possible, and find that the positive impact of deep trade agreements on GVC integration is driven by value added trade in intermediate rather than in final goods and services. Different from Laget et al. [62], scholars [61,63] use the EORA multi-regional input–output (MRIO) data to derive variables for trade in value added, as these offer greater country coverage. They found that broader PTAs have a larger impact on trade flows involving intermediates relative to flows involving all products, suggesting that GVC trade is particularly sensitive to the scope of trade policy cooperation [61]. However, they measure the coverage index of PTAs based on the number of provisions, we still don’t know whether the products mentioned in the PTAs promote the cooperation in the global production network. With the development of global production network and the shortage in the supply of some products under COVID-19 pandemic, a product-based measure for coverage of PTAs is needed to study the heterogeneity of PTAs and how this affect bilateral trade structure.

2: For me one of the major contributions is the construction of your original variable RTA_ratio. As such, I think you need to motivate with this more and make very clear what this does that is different from previous studies. What does this measure tell us that other measures have not? How is different from or similar to the DESTA depth variables? I think you could make the argument that it similar conceptually to depth but is measured in a different way (which is a unique contribution). Or, alternatively, you could make the argument that it is different in important ways to depth (which is more about the specificity of commitments related to non-trade aspects of PTAs), and this helps us better understand the effect of PTAs on trade flows. But this needs to be clear.

Then, in the empirics, you then need to compare your variable to others, such as the depth variable. For example, are these two measures correlated and, if so, how? Depending on whether you view them as complementary or overlapping this will determine, for example, whether you should include them in the same empirical models. Right now, you have them in the same econometric models later in the paper, but if they are substitute measures, they should likely not be included for conceptual and empirical reasons (multicollinearity).

Response: Again, thank you for the valuable comments and suggestions. This is actually the most important contribution of this paper. Although recently scholars start to pay attention to the heterogeneity of PTAs by analyzing the content of the documents. However, most of them focus on other aspects of PTAs other than the product market liberalization level. With the increasing development of global production network and the shortage of product supply under the COVID-19 pandemic, we think the product-based coverage index of PTAs are valuable to understand the heterogeneity of PTAs and their effect on bilateral trade. In this version, we rearrange the structure of our paper and add the construction of the product-based coverage index of PTAs section.

The introduction section shows the difference between this study and the previous literature: 

Countries are permitted to enter into PTAs under specific conditions covering trade in goods (Article XXIV of the General Agreement on Tariffs and Trade 1994), regional or global arrangements for trade in goods between developing country members (Enabling Clause), as well as agreements covering trade in services (Article V of the General Agreement on Trade in Services). Preferential trade agreements (PTAs) are an exception to the non-discrimination principle of the WTO, because only their signatories enjoy more favourable market-access conditions. Thus, different PTAs are heterogeneous in terms of the market access level and the cooperation that goes beyond tariff reductions, in services trade, investments, standards, public procurement, competition and intellectual property rights [2-4]. Previous studies have constructed variables to measure the heterogeneity of PTAs according to the content of PTAs document [5-7]. They construct depth variable to capture cooperation in areas such as services trade, investments, standards, public procurement, competition and intellectual property rights instead of product market liberalization of PTAs and flexibility variable to measure whether the PTAs allow states to temporarily withdraw concessions. However, with global value chains (GVCs) playing increasingly significant roles in the production of goods [11] and the shortage in the supply of some products under COVID-19 pandemic [12], measuring the product coverage of PTAs and identifying the effect of the product-based market liberalization of PTAs on bilateral trade are important. Some scholars have tried to analyze schedules of commitments in services PTAs by looking at specific market access and national treatment commitments in the 155 sub-sectors of the Services Sectoral Classification List (MTN.GNS/W/120) [8-10]. However, few studies take Goods and Services together into account and construct a product-based measurement to reveal market liberalization level of PTAs.

More scholars have studied how PTAs affect bilateral trade and concluded that PTAs can influence bilateral trade through trade creation effects and diversion effects [13-16]. Trade creation effects occurs when (because of relative efficiency) domestic production of less efficient member countries displaced by the imports from more efficient member countries. While trade diversion takes place when (because of preferences), low-cost non-member country’s imports are displaced by high-cost member country’s imports. Scholars established that the trade creation (increased trade because of relative efficiency) impact could higher compared to the trade diversion effect (increased trade because of preference) [17-18]. Although some scholar’s holds an opinion that PTAs are not consistent with an important principle of the multinational trading system and leads to unfair trade practices of trade disciplines [19-24]. PTAs have become a very appealing aspect of the international trading system [25] and according to (World Bank, 2018) [26], PTAs are currently the center of many policy-makers and scholars. Broadly speaking, PTAs pave the path to achieve “deep” economic integration by eliminating tariff. As mentioned before, PTAs are flexible and the product-based coverage of PTAs may differ from each other significantly. Most of existing studies referring to the effect of PTAs on bilateral trade mostly use dummy variable to measure whether countries share PTAs [27-29], a few scholars use the depth variable and found that the depth of PTAs matters for bilateral trade [5]. When decomposing bilateral trade into different components, although scholars have investigated the effect of PTAs on intermediate trade from global value chain (GVC) perspective and found that PTAs can facilitate supply chain activity, they mainly focus on the average effect of PTAs on GVC (using dummy variable to measure PTAs) and effect of provisions related to services, investment, and competition on GVC [30-32], starting from the product-based coverage of PTAs and test its effect on different types of trade is meaningful under the global production network background. Furthermore, some scholars tried to identify the heterogeneous effect of different PTAs on trade development and find that different PTAs can have different effect on bilateral trade [33], while most studies focus on the average effect of PTAs and knowledge of effect of PTAs on trade among different types of members are not enough. To conclude, constructing the product-based coverage of PTAs and investigate the effect of the product-based coverage of PTAs on different types of bilateral trade is important and scarce in existing literature.

We add a section to introduce our product-based coverage index of PTAs:

 3.2 Construction of the product-based coverage index of PTAs

We follow scholars who study the breadth of PTAs [10, 59-61] to construct a coverage index of PTAs based on the content of PTAs. We focus on the product-based coverage of PTAs instead of coverage of provisions referring to other aspects, while existing literature focus on the industry-based coverage index of service PTAs [10] or a count of provisions covered by a PTA [59-61]. Based on WTO rules on PTAs, PTAs are under specific conditions covering trade in goods (Article XXIV of the General Agreement on Tariffs and Trade 1994), regional or global arrangements for trade in goods between developing country members (Enabling Clause), as well as agreements covering trade in services (Article V of the General Agreement on Trade in Services). We firstly download all PTA documents (302 in total) manually on WTO website extract all the products whose trade barriers will be removed to promote free trade . To calculate a comparable measure for all PTAs, we match the products mentioned in each PTA document with the ISIC Rev.4, which includes 56 industries (1 to 23 are agricultural and manufacturing goods industries, 24 to 56 are service industries). We identify whether the RTA cover this industry by searching all products in goods industries and keywords in service industries. For example, China and South Korean signed a PTA which was in force in late 2015, we search the document which is in Chinese, and it mentioned products like fish, wood, bank service, etc. Based on the ISIC Rev.4, we match fish to “3. Fishing and aquaculture”, wood to “7. Manufacture of wood and of products of wood and cork, except furniture; manufacture of articles of straw and plaiting materials”, bank service to “43. Activities auxiliary to financial services and insurance activities” respectively. The coverage index of PTAs is calculated based on equation (8), where Coveredi =1 if the RTA document covers the products of industry “i”, otherwise Coveredi =0. This is similar to the count of provisions covered by a PTA [59-61], however, we differ from them by using the ratio of covered industries based on the products. 

 （8）

We take China as an example to verify our product-based coverage index of PTAs , which is shown in Figure 2. China's regional trade agreements since 2000 show the characteristics of differentiation: China-Switzerland free trade agreement and China-Iceland free trade agreement, which came into effect on July 1, 2014, have a broad coverage in terms of trade in goods and trade in services. Switzerland and Iceland, as the most important non EU countries in Europe, are China’s important economic and trade partners in Europe. The two agreements have wide coverage, high level of openness, and many preferential policies. They are high-quality, wide-ranging, mutually beneficial and win-win free trade agreements. They are also one of the highest level and most comprehensive free trade agreements reached by China in recent years. The product-based coverage index of this paper shows that the coverage rate of goods trade and service trade of China-Switzerland free trade agreement is 100% and nearly 90% respectively, which has a strong fit with the official data. The official statement of the China-Georgia free trade agreement states: "Georgia will immediately implement zero tariff on 96.5% of China's products, covering 99.6% of Georgia's total imports from China; China will implement zero tariff on 93.9% of Georgia's products, covering 93.8% of China's total imports from Georgia, including 90.9% of products (42.7% of total imports) immediately implementing zero tariff, and the remaining 3% of products (51.1% of total imports) The transitional period of tax reduction is five years. In the field of trade in services, the two sides have made high-quality open commitments to many service sectors. Among them, Georgia has met China's key concerns in the fields of finance, transportation, movement of natural persons, and traditional Chinese medicine services, while China has met Georgia's key concerns in the fields of tourism, shipping, and law. " The coverage index of this paper shows that the coverage rate of goods trade and service trade of China Georgia free trade agreement is 96.56% and 24.24%, which proves the accuracy of this product-based coverage index. The product-based coverage index of other PTAs referring to China shown in Figure 2 are also consistent with the description of China's regional trade agreements, which proves the accuracy of this measure.

Figure 2 the product-based coverage index of PTAs referring to China 

We then show the trend of product-based coverage index of PTAs in Fig 3. It is evident from the Figure that product-based coverage index of PTAs is increasing since the 21st century. However, it is also shown in this figure that the variance of this trend is large, we calculate the mean value of product-based coverage index of all PTAs based on the year in force, this could affect the trend due to the time lag. Also, when we look deeply to the components, we find that the coverage index of service industries of PTAs is growing more significantly than the goods industries. Overall, the upward trend can be concluded if we focus on the whole picture. 

Figure 3 The trend of coverage ratio index of PTAs over the years 2000 to 2019. The documents of PTAs are from WTO website: http://rtais.wto.org/UI/PublicMaintainRTAHome.aspx. We calculate the mean value of coverage index of PTAs in each year and we can see that the trend is upward.

We need to mention that in the empirical regression, we need to construct country-pair observations. In multilateral agreements, a country and an economic organization divided into a country and the economic organization that shares the same RTA, for example, ASEAN. For any two countries in the ASEAN, they share the same PTAs and the coverage index of RTA. In 2002, CAFTA (China and ASEAN Free Trade Area) signed by China and ASEAN countries. In this case, China and each country of ASEAN share the same RTA and the coverage index of RTA. To conclude, if an agreement is sign within an economic organization, all countries included in the economic organization would be match with each other, and finally multiple countries will share the agreement. Similarly, we treat other multilateral agreements and get the reciprocal industry penetration of agreements countries. Based on the coverage index of PTAs, this paper studies the relationship between PTAs and bilateral trade.

We then compare the product-based coverage index of RTAs with other depth measures, we prove that there is no multicollinearity threat:

To verify our results, we also add other measures of PTAs and compare them with the product-based coverage in our analysis. Specifically, we added the depth and flexibility variables in our empirical model to control the cooperation that goes beyond tariff reductions, in services trade, investments, standards, public procurement, competition and intellectual property rights . Dür et al. [5] used two different measures to operationalize the depth of PTAs. The first measure of depth is an additive index that combines seven key provisions that can be included in PTAs, and the second one relies on latent trait analysis (depth_rasch) [5]. Baccini et al. [6] measures the flexibility (flexescape) of PTAs based on the content and find that positive relationship between depth and flexibility holds for preferential trade agreements (PTAs). We compare the product-based coverage index of PTAs with the depth and flexibility measure and the results are shown in Table 1. We can conclude that the product-based coverage index of PTAs is positively correlated with the RTA dummy, depth of PTAs and flexibility of PTAs, and the correlation coefficients are 0.749, 0.327, 0.65 respectively, suggesting that there is no collinearity problem between PTA_ratio, depth_rasch and flexescape.

Table 1.

Correlation matrix of different measures of PTAs 

 PTA PTA_ratio depth_rasch flexescape

PTA 1.000 

PTA_ratio 0.749*** 1.000 

depth_rasch -0.141*** 0.327*** 1.000 

flexescape 0.892*** 0.650*** -0.093*** 1.000

Note:p-values in brackets * p < 0.1, ** p < 0.05, *** p < 0.01. 

3: Another contribution seems to be that you are using a different dependent variable for some of your regressions, but you don’t really sell this that effectively. If this is novel, which I believe it is, then your paper is making an important contribution that RTAs with greater scope of products covered don’t just increase trade in general they especially increase trade in intermediate goods, which are important for supply chains. This is a good contribution, but it is hidden in the manuscript.

Response: Thank you for the valuable comments and suggestions. With the increasing development of global production network and the shortage of product supply under the COVID-19 pandemic, the product-based coverage index of PTAs may affect bilateral trade in different ways. We follow existing literature referring to global value chain (GVC) and decompose into different components. We clarify this contribution in the introduction section, literature section, and the empirical section.

The introduction section:

When decomposing bilateral trade into different components, although scholars have investigated the effect of PTAs on intermediate trade from global value chain (GVC) perspective and found that PTAs can facilitate supply chain activity, they mainly focus on the average effect of PTAs on GVC (using dummy variable to measure PTAs) and effect of provisions related to services, investment, and competition on GVC [30-32], starting from the product-based coverage of PTAs and test its effect on different types of trade is meaningful under the global production network background. Furthermore, some scholars tried to identify the heterogeneous effect of different PTAs on trade development and find that different PTAs can have different effect on bilateral trade [33], while most studies focus on the average effect of PTAs and knowledge of effect of PTAs on trade among different types of members are not enough. To conclude, constructing the product-based coverage of PTAs and investigate the effect of the product-based coverage of PTAs on different types of bilateral trade is important and scarce in existing literature.

The literature review section:

In addition to the effect of PTAs on bilateral trade and the design of PTAs, this paper also investigates how PTAs affect intermediate good trade. Existing literature referring to global value chain (GVC) tend to study how deep trade agreements affect different type of trade recently [61-63]. Laget et al. [62] use trade data in both value-added and gross form to provide as complete a picture of GVC trade as possible, and find that the positive impact of deep trade agreements on GVC integration is driven by value added trade in intermediate rather than in final goods and services. Different from Laget et al. [62], scholars [61,63] use the EORA multi-regional input–output (MRIO) data to derive variables for trade in value added, as these offer greater country coverage. They found that broader PTAs have a larger impact on trade flows involving intermediates relative to flows involving all products, suggesting that GVC trade is particularly sensitive to the scope of trade policy cooperation [61]. However, they measure the coverage index of PTAs based on the number of provisions, we still don’t know whether the products mentioned in the PTAs promote the cooperation in the global production network. With the development of global production network and the shortage in the supply of some products under COVID-19 pandemic, a product-based measure for coverage of PTAs is needed to study the heterogeneity of PTAs and how this affect bilateral trade structure.

The empirical section:

With the increasing development of global value chains (GVCs) and the shortage in the supply of some products under COVID-19 pandemic, our measure based on the product covered in PTAs is important for studying the effect of PTAs on bilateral trade. Further, we use EORA input-output database to calculate bilateral trade volume and decompose total trade into components in terms of types of trade (intermediate goods, final goods; commodity and service trade). We investigate the product-based coverage index of PTAs on these different trade types. The empirical results given in Table 4. The findings of column (1) and (2) showed that PTAs with a higher degree of openness have a significant positive promotional impact on both intermediate goods trade and final goods trade. As the estimated coefficients are 0.0408 and 0.030, which means that when the product-based coverage of the PTAs is 1, the maximum promotion effect of the implementation of the PTAs on intermediate goods trade is 4.185%. The maximum promotion effect on the final product trade is 3.045%. PTAs cannot only help countries to sell final products and hence promote the welfare of countries, but also embedded in the global value chain through intermediate goods trade, suggesting that the product liberalization brought by PTAs can promote the development of global production network. Countries can sign trade agreements covering a wider range of products to help them better integrate into the global value chain and promote their own trade development level. Further, findings of the differential impact of PTA_ratio on service trade and final goods trade are reported in column (3) and (4) of Table 5. Results showed that regional service trade agreements have a significant promotion effect on trade flow, with coefficients of 0.0436 and 0.0411, respectively. It implies that the maximum promotion effect of the implementation of PTAs on services trade is 5.527%, and the maximum promotion effect on trade goods is 4.185%. PTAs are not only covering tariff barriers, but also the ‘new age’ matters such as foreign direct investment, services, labor and environmental standards. The service trade promotion of PTAs shows that the high-standard PTAs are helpful for countries to open up more broadly and improve the overall trade status.

Table 5

Empirical results of the coverage index (PTA_ratio) on trade components 

Variables lnmid_trade（1） lnfd_trade

（2） lnservice

（3） lncommo

（4）

PTA_ratio 0.0387*** 0.0509*** 0.0538*** 0.0334***

 (0.009) (0.010) (0.011) (0.008)

depth_rasch 0.0313*** 0.0263*** 0.0420*** 0.0153***

 (0.003) (0.004) (0.004) (0.003)

flexescape -0.0020 -0.0043** -0.0062*** 0.0025*

 (0.002) (0.002) (0.002) (0.001)

Comcur 0.0210** 0.0282** 0.0232** -0.0036

 (0.009) (0.011) (0.012) (0.007)

Country-pair FE Y Y Y Y

Country-year FE Y Y Y Y

Obs 721,135 721,135 721,135 721,135

R2 0.9966 0.9961 0.9955 0.9975

Note: See note under Table 3

4: A final contribution is the finding that perhaps the trade creating effects of wider scope PTAs are most evident in developing country dyads. This is an important contribution too, but the writing here obscures the point. Make this clear. Also, try to make the language consistent with previous efforts.

Response: Thank you for the valuable comments and suggestions. This is also an important contribution of this study. We divide sample into different groups based on the economic development level of member countries of PTAs, and we compare the effect of different measures on bilateral trade (we have proven that there is no muticollinearity threat in the correlation matrix), the results are interesting and we find that the trade creating effects of wider scope PTAs are most evident in developing country dyads. We revise this part as follows:

Scholars have constructed other variables to measure the heterogeneity of PTAs according to the content of PTAs [22-24]. We add these variables in our model and the empirical results show that the coverage index of PTAs has significant positive effect on bilateral trade in the full sample, which shown in column (1) of Table 3. We also find that the depth of PTAs can promote bilateral trade as well, but the marginal effect (0.0316) is less than the coverage index (0.0450). Moreover, the flexibility of PTAs has a negative effect on bilateral trade, and the reason behind this is that the PTAs are flexible to adjust their policies for other purposes without violating the terms of an agreement. They could not cut the tariff as they promise because they can withdraw the PTAs. This is important for countries to design flexible PTAs. Flexible PTAs provide for legally accepted opt-outs without leading to a de jure breach of an agreement and encompass exit options, duration and renegotiation clauses, reservations, escape clauses, and withdrawal clauses, and hence help members to cooperate better [23]. The product-based coverage of PTAs differs them in focusing on the market access openness of PTAs instead of other aspects of PTAs.

We further analyzed the market access effect of PTAs. We used World Development Indicators (WDI) to identify whether a country is High-income and then we divided the full sample into different groups. The result presented in column (2) - (4) of Table 4. The results of column (2) and (3) showed that the coverage index of PTAs is insignificant if the country is from high-income group. However, the depth of PTAs is still significant at 1% level. When both countries are not high-income countries, the coverage index of PTAs is significant at 1% level. Based on results we concluded that the product-based coverage index of PTAs could promote bilateral trade both for middle-income or low-income groups, because, to liberalize market access barriers is crucial for these countries. However, for the high-income countries, the market access barriers are already less and the depth of PTAs is much more important to promote bilateral trade. The results shed light on the globalization development for different countries. For the high-income countries, they should focus on other aspects of PTAs which can contribute to higher quality trade. However, for the less developed countries, they should liberalize the product market gradually to promote bilateral trade.

Table 4

 Robustness checks of the product-based coverage index (PTA_ratio) on bilateral trade 

Variables Full Sample

(1) High-High

（2） High-No High（3） No High-No High

(4)

PTA_ratio 0.0450*** 0.0173 0.0163 0.0849***

 (0.009) (0.016) (0.021) (0.015)

depth_rasch 0.0316*** 0.0375*** 0.0341*** 0.0199***

 (0.004) (0.005) (0.010) (0.006)

flexescape -0.0031* -0.0098*** -0.0070 -0.0009

 (0.002) (0.003) (0.004) (0.003)

Country-pair FE Y Y Y Y

Country-year FE Y Y Y Y

Obs 721,135 80,252 166,971 474,912

R2 0.9965 0.9985 0.9980 0.9959

Note: we divide our full sample into different samples based on whether the country is high-income group. High- High denotes both countries are of high income，High- No high means one country is of high income and the other is of middle or low income. No high -No high means both countries are from middle or income group.

5: Finally, I don’t really understand the rationale for the instrumental variable regressions. What type of endogeneity are you concerned with, and how is your selected IV going to address it? You mention the inclusion of democracy and other “political” variables in other papers, but these are often used as a control variable not in a 2 stage model. Moreover, the Baccini et al (2015) paper you cite is using joint democracy: A) as a control and B) in econometric models where the dependent variable is flexibility (from DESTA), not trade volumes. Overall, if you want to keep the IV regression, much more would need to be done to explain what value added it provides and then you need to show that the IV you select satisfies the exclusion restriction requirements.

Response: Thank you for the valuable comments and suggestions. We totally agree with you on this and we delete the 2SLS part of this study. 

6: You never really discuss what the acronym EORA stands for or explain what this data is. I had to look it up.

Response: Sorry that we don’t make this clear and thank you for this suggestion. We add the construction of bilateral trade section to introduce this carefully:

3.3 Construction of bilateral trade variables and other variables 

Following previous studies [61-62], We use the latest release of the EORA multi-regional input–output (MIRO) tables [71]. The rows in an MRIO table indicate the use of gross output from a particular industry in a particular country and comprise two main components. The first is intermediate use, which provides information on intermediate use by both domestic industries and industries in other countries. The second is information on final demand, which is again split between

---

## [Decision Letter · Decision Letter 2]

17 Feb 2021

PONE-D-20-21416R2

The impact of Preferential Trade Agreements on Bilateral Trade: A Structural Gravity Model Analysis

PLOS ONE

Dear Dr. zhang,

Thank you for submitting your manuscript to PLOS ONE. After careful consideration, we feel that it has merit but does not fully meet PLOS ONE’s publication criteria as it currently stands. Therefore, we invite you to submit a revised version of the manuscript that addresses the points raised during the review process.

The reviewer raises excellent points and even provides suggestions for how to improve the presentation of the paper. I will very carefully check the revised paper for whether you have satisfactorily addressed all the reviewer's comments.  Please also indicate a link to replication materials so that researchers can benefit from your new measure.

We look forward to receiving your revised manuscript.

Kind regards,

Bernhard Reinsberg, Ph.D

Academic Editor

PLOS ONE

Reviewers' comments:

Reviewer's Responses to Questions

**Comments to the Author**

1. If the authors have adequately addressed your comments raised in a previous round of review and you feel that this manuscript is now acceptable for publication, you may indicate that here to bypass the “Comments to the Author” section, enter your conflict of interest statement in the “Confidential to Editor” section, and submit your "Accept" recommendation.

Reviewer #3: (No Response)

2. Is the manuscript technically sound, and do the data support the conclusions?

Reviewer #3: Yes

3. Has the statistical analysis been performed appropriately and rigorously? 

Reviewer #3: Yes

4. Have the authors made all data underlying the findings in their manuscript fully available?

Reviewer #3: No

5. Is the manuscript presented in an intelligible fashion and written in standard English?

Reviewer #3: No

6. Review Comments to the Author

Reviewer #3: Overall, I think the manuscript is improved over the first version. In particular, the argument and writing is easier to follow on the whole. Moreover, I can see the paper being of interest to readers.

That said, the quality of argumentation and presentation of results in the manuscript is still lacking. As such, I would not recommend publication until further revisions have been made. And I will stress this point – the quality of the argumentation and writing needs to be upgraded all around. As such, the authors would need to go through the manuscript very carefully to improve it by making it more readable, consistent, and direct; simply adding/improving some sections is insufficient on its own.

Here are some more specific suggestions:

1) the abstract doesn’t do a great job motivating the study. In fact, it begins with two somewhat disputed claims (that PTAs increase trade and that structural gravity models are the best way to study). I would change this so that you are motiving the study around a debate/question or a contribution. So, for instance you could say “trade agreements are thought to raise trade-integration…… but existing measures are limited…” or something to this effect. Or alternatively you could just start by directly stating what your paper contributes. I would read papers with good abstracts to see what the best ones look like.

2) Make sure to change articles (a, an, and the) from definite to indefinite when the identity of the noun is unknow. For example, in the first sentence it should be “we develop a product-based coverage index..” rather than “we develop the product-based coverage index…”.

3) I am not sure that motivating the global value chain (GVC) aspect of the study solely with respect to COVID-19 is the right way to go (you aren't testing anything related to COVID-19). Yes, the pandemic makes it more important potentially, but I think it is an important dynamic to study even without the pandemic, as GVCs have increased in importance in recent years.

4) You still regularly do not spell out acronyms the first time you use them and/or you spell them out multiple times later, which you don’t need to do. For instance, EORA is used in the abstract without defining, and ISIC is used this way also. But, in other parts, you regularly spell out PTAs each time it is used.

These little issues can add up in the eyes of a reviewer. You need to go through the manuscript very carefully to minimize these small issue.

There is also a point in the manuscript that says “However, Moreover….”

5) You do a better job in several places selling your product-based coverage measure, but the writing is still somewhat indirect. Make sure to state very clearly that previous models don’t have a fine-grained measure of market access (unless they do, and you should make this clear). This is one of the selling points of the paper and the prim value added. Explain very directly why we need this and what it adds to the literature on trade agreements!

6) The example of your PTA_ratio variable and Chinese PTAs are good and illustrative. Two minor things though: 1) not sure if Iceland is one of the two most important non-EU countries (Britain, Norway,?); and 2) you have some pejorative sounding statements. For example, you say “mutually beneficial and win-win free trade agreements” when describing China’s PTAs -- try to keep language more neutral. Also, slightly confusing when you pivot from talking about China-Switzerland to China-Georgia.

7) Table 1. I am not sure why you put this here and not perhaps before table 4 where it is more applicable (but in general its location is not as important as explaining why you did it). Also, I am not sure why we need to know the correlation between the PTA dummy and the other 3 variables (you are correlating a binary variable and continuous variables)? Second, you state that multicollinearity is not a problem but your correlation coefficient between PTA ratio and flexscape is .65. This is pretty high. Better to check the variance inflation factor (VIF) between your independent variables and then, if the VIF is too high, take corrective measures. Also, of note, is that the correlation between you PTA ratio and depth is only .327. I would discuss this. I think it potentially makes your paper stronger by showing that you are measuring something that depth doesn’t capture.

Relatedly, some of the walk though of your descriptive stats is clunky in this section of the paper. For example, you say that “While the regional trade dummies show, more average probably 15.5%”. This just means that 15.5 % of your dyads have a PTA in a given year, correct?

7) Label the models in table 1.

8) In the paragraph below table 3 you note that you have included “other variables to measure the heterogeneity of PTAs” and then say that you included depth and flexibility in model 1. But this is very confusing: 1) model 1 in table 3 does not include your product-based coverage measure, and 2) model 1 doesn’t seem to include the other depth variables (are they estimated but not included in table 3). This also makes the correlations you presented in table 1 more confusing. What is the purpose now? In general, much more attention to detail needs to be taken in walking through all this. The results become confusing unless you are very careful, and this then makes the reader more and more skeptical.

9) Table 4. Explain why you decided to do this? I find the endeavor interesting and worthwhile, but it comes across as a bit ad-hoc. Provide some rationale. Also, maybe you don’t want to call it “robustness” checks. I think really what you are doing is showing how your measure is different from and complements depth and flexibility, frame it as such.

Also, it seems that table 4 is where you are including depth and flexibility, not table 3. Make sure this is clear. Also, from a theoretical and empirical standpoint why are you including here and not in table 3? Why are you not including same controls etc? These decisions should be transparent and motivated (you can footnote these justifications as necessary, but the reader shouldn’t have to speculate). I am not saying that any of these decisions are wrong or need to be changed. Rather, you just need to be more transparent and deliberate about communicating to the reader what you are doing and why.

10) Your note for table 5 just says “see note under Table 3. I don’t think the notes are or should be the same…

7. PLOS authors have the option to publish the peer review history of their article (what does this mean?). If published, this will include your full peer review and any attached files.

Reviewer #3: No

---

## [Author Response · Author response to Decision Letter 2]

10 Mar 2021

Reply to Reviewers Comments for Paper PONE-D-20-21416 R2

“The impact of Preferential Trade Agreements on Bilateral Trade: A Structural Gravity Model Analysis”

Dear Editor and Reviewers,

We would like to commence by thanking the editor and the reviewers for their valuable time and constructive comments. Their expert knowledge of the field has helped us to strengthen the manuscript significantly. We very much appreciate Reviewer#3’s valuable suggestions, which is really helpful for us to revise the manuscript. We endeavored to address all the comments and our reflections are given below point by point. We have taken English service to make our paper more readable.

To help researchers benefit from our new measure, we upload our measure to the Journal website. 

Sincerely,

The Authors

Response to Reviewer #3’s Comments

Reviewer #3: 

Reviewers' general comments:

Overall, I think the manuscript is improved over the first version. In particular, the argument and writing is easier to follow on the whole. Moreover, I can see the paper being of interest to readers. 

Response: Thank you very much for your valuable suggestions. We are grateful to your valuable suggestions, which is helpful for us to revise the manuscript. In this new version, we have taken English service to improve the language of the entire manuscript, and taken your advice to adjust the empirical section. Hopefully we can meet your demand and help you understand our contributions to existing literature. Again, thank you very much for your valuable suggestions.

Specific comments:

1: the abstract doesn’t do a great job motivating the study. In fact, it begins with two somewhat disputed claims (that PTAs increase trade and that structural gravity models are the best way to study). I would change this so that you are motiving the study around a debate/question or a contribution. So, for instance you could say “trade agreements are thought to raise trade-integration…… but existing measures are limited…” or something to this effect. Or alternatively you could just start by directly stating what your paper contributes. I would read papers with good abstracts to see what the best ones look like.

Response: Thank you for the valuable comments and suggestions. We take your advice and revise the abstract as follows:

Trade agreements are thought to raise trade integration, but existing preferential trade agreements (PTAs) are insufficient in measuring market access of products. This study develops a product-based coverage index of PTAs using the World Trade Organization (WTO) preferential trade agreements and calculates bilateral trade measures using the EORA multi-regional input-output (MRIO) tables covering 189 countries worldwide over the period 1990–2015; the structural gravity model is employed to test how PTAs affect bilateral trade. Our findings show that countries sharing a common PTA could boost the trade volume compared to those without PTAs, supporting the trade creation effect. However, the trade promotion effect of the product-based coverage index of PTAs is significant only if the member countries are low-and middle-income countries. Further, the wide range of product liberalization brought by PTAs can promote global production networks by stimulating the trade of intermediate goods. Our results are important for understanding the market access effect of PTAs with the increasing development of trade integration and global value chains (GVCs).

2: Make sure to change articles (a, an, and the) from definite to indefinite when the identity of the noun is unknow. For example, in the first sentence it should be “we develop a product-based coverage index.” rather than “we develop the product-based coverage index…”.

Response: Thank you very much for your careful reading and valuable comments and suggestions. We read the whole carefully and take English service to smooth the whole paper this time.

3: I am not sure that motivating the global value chain (GVC) aspect of the study solely with respect to COVID-19 is the right way to go (you aren't testing anything related to COVID-19). Yes, the pandemic makes it more important potentially, but I think it is an important dynamic to study even without the pandemic, as GVCs have increased in importance in recent years.

Response: Thank you for the valuable comments and suggestions. We motivate the global value chain without referring to COVID-19 in this version. 

4: You still regularly do not spell out acronyms the first time you use them and/or you spell them out multiple times later, which you don’t need to do. For instance, EORA is used in the abstract without defining, and ISIC is used this way also. But, in other parts, you regularly spell out PTAs each time it is used.

Response: Thank you very much for your careful reading and valuable comments and suggestions. We read the whole carefully and take English service to smooth the whole paper this time. We try to spell out acronyms the first time we use them and use them after that. Sorry for our careless about this.

5: You do a better job in several places selling your product-based coverage measure, but the writing is still somewhat indirect. Make sure to state very clearly that previous models don’t have a fine-grained measure of market access (unless they do, and you should make this clear). This is one of the selling points of the paper and the prim value added. Explain very directly why we need this and what it adds to the literature on trade agreements!

Response: Again, we very much appreciate your kind work. The product-based coverage measure is one of our contribution to the related literature, and we stress the reason why we construct this measure in multiple places, especially in the literature review section. Existing literature referring to depth of PTAs focus on provisions beyond tariff, such as services, investment, standards, intellectual property, and competition rules, as well as a host of issues not directly related to trading, such as the environment. Some scholars measure the market access level brought by PTAs, while they mainly focus on service trade and calculate the index on service industry level. We consider commodities and service together and construct the product-based coverage index. We further investigate the trade effect of the product-based coverage index of PTAs. The literature review are as follows:

2 Literature review 

In this section, we review the related literature to clarify our contribution from the literature perspective. We mainly focus on two strands of literature: the effect of PTAs on bilateral trade and the design of PTAs.

PTAs have been proliferating for the last twenty years. A large body of literature has studied various aspects of this phenomenon and study its design. Some researchers have highlighted that bilateral trade agreements are obstacles to free trade [34]. Regarding the obstacle [35–39], researchers argued that PTAs undermine multilateral growth and unleash a protectionist spiral. They further stated that PTAs effects like the spaghetti bowl effect and reduced countries’ incentive to enter multilateral trade agreements. Regionalism leads to welfare losses in both member and excluded countries [40]. In a Ministerial Meeting of the WTO in Doha (2001), governments expressed their opinion that “regional trade agreements can play an important role in promoting the liberalization and expansion of trade” [41]. The motivation for this evolution in PTAs is to ease tariff barriers and ‘new age’ matters such as foreign direct investment, services, labor, and environmental standards [28]. More scholars have applied the gravity model to estimate the effect of PTAs and find a positive effect on bilateral trade [39, 40]. However, most studies conceptualize PTAs as a dichotomous variable, namely whether countries sign an agreement or not and hence treat PTAs as if they were all equal in purpose while estimating the effects of PTAs. The results are fruitful, and they conclude that, on average, PTAs can promote bilateral trade for member countries. Some scholars estimate that the long-run effect of PTAs on bilateral trade flows is 100% and the effect varies considerably across trade agreements [42, 43]. Others applied the structural gravity model to analyze the PTAs and concluded that the average treatment effect of PTAs on trade flows was 236% [44]. Dembatapitiya and Weerahewa [45] investigated the SAFTA, European Union (EU), ASEAN, BIMSTEC, and North American Free Trade Agreement (NAFTA) regional trade impact and found mixed results. For example, the co-efficient for the EU is significant, while SAFTA, ASEAN, BIMSTEC and NAFTA do not significantly impact bilateral trade. Sampson [46] studied on the evolution of China’s PTAs and stated that the increasing network of China’s PTAs is important and strategic for the Asian region. Pant and Paul [47] evaluated PTAs for India and argued that PTAs are good for the intra-regional trade volume and welfare of countries. Scholars also assess the ex-post trade effects by applying the gravity model [48, 49]. Carre`re [50] studied ex-post PTAs and claimed that intra-regional trade mostly tied to a reduction in imports. However, regional agreements back weak governments to implement reforms and stabilize them despite domestic opposition. Partners may learn the advantages of liberalization once they practice limited free trade.

Until recently, scholars investigated the content and design of PTAs [51] and studied regional specifications [52] or explained functional differences in design, for example, dispute settlement [53] and flexibility provisions [54]. Furthermore, they found that tariff cuts, and other market access and trade-related provisions in PTAs concerning topics such as investments and intellectual property rights matter for trade flows [5], and that such deep agreements are usually flexible in adjusting their policies for other purposes and withdraw the PTAs without violating the terms of an agreement [6]. In addition to market access to goods, many PTAs today include provisions in trade disciplines such as services, investment, standards, intellectual property, and competition rules, as well as a host of issues not directly related to trading, such as the environment. At the beginning of the study of PTAs’ design, scholars usually focus on specific PTAs and measure the strength of a wide variety of provisions in the legal texts of PTAs [10, 55]. With increasing number of documents on PTAs available on the WTO website, scholars have extended their objects and started to pay greater attention to the scope and depth of these agreements [5–10,55–58]. They focus on the broader economic integration rights in goods, services, and factor markets brought by PTAs. For example, Dür et al. [5] used two different measures to operationalize the depth of PTAs. The first measure of depth is an additive index that combines seven key provisions included in PTAs, and the second one relies on latent trait analysis. Baccini et al. [6] measured the flexibility of PTAs and found a positive relationship between depth and flexibility for PTAs. Hofmann et al. [7] offered a detailed assessment of preferential arrangement. They examined the coverage and legal enforceability of provisions regulating a large set of policy areas. In addition to the depth of PTAs, many scholars have studied the breadth of PTAs [59–61]. Miroudot et al. [10] attempted to construct an industry-based coverage index of PTAs; however, they only studied 56 services PTAs. Limão claims that a broader PTA is one where partners seek to increase market access in product markets, and in markets for capital, labor, and technology [59]. Based on this, Hofmann et al. construct the measure based on a count of the provisions covered by a PTA and can focus either on the 18 ‘core’ provisions [60] and recent scholars also build a measure following this method [61]. Although some scholars have constructed variables to measure the heterogeneity of PTAs, most of them focus on the non-trade aspects of PTAs, and few measure the coverage index from the product-based perspective, which is important in the context of global production network. 

In addition to the effect of PTAs on bilateral trade and the design of PTAs, this study also investigates how PTAs affect intermediate goods trade. Existing literature referring to the GVCs tends to study how deep trade agreements affect different trade types [61–63]. Laget et al. [62] use trade data in both value-added and gross forms to provide a complete picture of GVCs trade as possible and found that the positive impact of deep trade agreements on GVCs integration is driven by value-added trade in intermediate rather than final goods and services. In contrast to Laget et al. [62], scholars [61, 63] use the EORA multi-regional input–output (MRIO) data to derive variables for trade in value-added, as these offer greater country coverage. They found that broader PTAs have a larger impact on trade flows involving intermediates relative to flows involving all products, suggesting that GVCs trade is particularly sensitive to the scope of trade policy cooperation [61]. However, they measure the coverage index of PTAs based on the number of provisions, and we still do not know whether the products mentioned in the PTAs promote cooperation in the global production network. With the development of global production network, a product-based measure for the coverage of PTAs is needed to study the heterogeneity of PTAs and how this affects bilateral trade structure.

6: The example of your PTA_ratio variable and Chinese PTAs are good and illustrative. Two minor things though: 1) not sure if Iceland is one of the two most important non-EU countries (Britain, Norway,?); and 2) you have some pejorative sounding statements. For example, you say “mutually beneficial and win-win free trade agreements” when describing China’s PTAs -- try to keep language more neutral. Also, slightly confusing when you pivot from talking about China-Switzerland to China-Georgia.

Response: Again, we very much appreciate your kind work. In last version, we try to list as many PTAs signed by China as possible to verify our measure, and the results fit the facts well. In this version, we keep the discussion of China- Switzerland and China-Iceland and delete the China-Georgia to make paper more readable. And the revision is as follows:

We take China as an example to verify our product-based coverage index of PTAs , which is shown in Fig 2. China's regional trade agreements since 2000 show the characteristics of differentiation: the China-Switzerland free trade agreement and China-Iceland free trade agreement, which came into effect on 1 July, 2014, have a broad coverage in terms of trade in goods and trade in services. As important non-EU countries, Switzerland and Iceland are China’s important economic and trade partners in Europe. The two agreements have wide coverage, high level of openness, and many preferential policies. They are high-quality and wide-ranging free trade agreements. They are also one of the highest level and most comprehensive free trade agreements reached by China in recent years. The product-based coverage index shows that the coverage rate of goods and service trade of China-Switzerland free trade agreement is 100% and nearly 90% respectively, which has a strong fit with the official data.

Fig 2 The product-based coverage index of PTAs referring to China

7: Table 1. I am not sure why you put this here and not perhaps before table 4 where it is more applicable (but in general its location is not as important as explaining why you did it). Also, I am not sure why we need to know the correlation between the PTA dummy and the other 3 variables (you are correlating a binary variable and continuous variables)? Second, you state that multicollinearity is not a problem but your correlation coefficient between PTA ratio and flexscape is .65. This is pretty high. Better to check the variance inflation factor (VIF) between your independent variables and then, if the VIF is too high, take corrective measures. Also, of note, is that the correlation between you PTA ratio and depth is only .327. I would discuss this. I think it potentially makes your paper stronger by showing that you are measuring something that depth doesn’t capture.

In the paragraph below table 3 you note that you have included “other variables to measure the heterogeneity of PTAs” and then say that you included depth and flexibility in model 1. But this is very confusing: 1) model 1 in table 3 does not include your product-based coverage measure, and 2) model 1 doesn’t seem to include the other depth variables (are they estimated but not included in table 3). This also makes the correlations you presented in table 1 more confusing. What is the purpose now? In general, much more attention to detail needs to be taken in walking through all this. The results become confusing unless you are very careful, and this then makes the reader more and more skeptical.

Response: Again, we very much appreciate your kind work. In this version, we make two distinct adjustment about this. First, 

In our baseline model, we do not include the depth and flexibility measure, because we want to show the basic effect of PTAs on bilateral trade with the factors controlled following most of existing studies. We include the depth and flexibility measure in the robustness checks to show that the results are robust taking other measure of PTAs into account. Before the robustness checks, we take your advice and put the correlation matrix table before table 4 (we delete the dummy variable in the correlation matrix table). Second, we check the variance inflation factor (VIF) and the values are all below 10, suggesting that there is no multicollinearity problem. The revisions are as follows:

4.1 Baseline results of how PTAs affects bilateral trade

First, we test the effects of PTAs on bilateral trade. The results of the panel structural gravity model are presented in Table 2. We first used the dummy variable (PTAs) to measure whether a trade agreement exists the two-countries, and the results are shown in column (1)–(3). The results in column (1) of Table 2 shows that the impact of PTAs on bilateral trade is positive and significant. This implies that a trade agreement helps increase the trade flow between the member countries of PTAs. These results are consistent with the existing study [75]. We also control for the economic scale effect. The economic scale effect (income-effect) of exporter/importer countries (agreement partner) on bilateral trade flow is significantly positive. Surely economic scale effect raises the bilateral trade between partners. It is remarkable to note that the magnitude impact (0.31 for importers, 0.35 for exporters) of both partners is significant at the 1% level but slightly different. This implies that trade will increase at a significant level on both sides which ultimately increases the welfare gains for both countries. The positive impact of income was validated with Yao et al. [76]. However, geographical distance negatively influences bilateral trade flow, consistent with the traditional gravity model results. This implies that distance causes a decrease in bilateral trade due to the shipping cost of transferring goods from one country to another. Therefore, adjacent countries promote bilateral trade rather than distant countries. Consistently, our results confirmed that the adjacent countries contribute positively to increasing bilateral trade, as the coefficient (0.3975) impact is significantly positive. Communication is important for promoting trade between two countries [77]. For example, each country has its rules and regulations, which are settled between the two countries through communication. Therefore, a common language can be a powerful tool to deal with these issues, that is, more communication in the same language, more understating more international supply chain and thereby more trade [76]. The colonial relationship also affects trade between two countries [77]. Our results also show the positive influence of colonial on trade follow. This means that the colonies strengthen the trade relationship. Currency helps to determine a country’s economic health. The same currency would help promote trade between countries as trade becomes cheaper if trading partners have the same currency [78]. In addition, our results showed a positive impact on trade flow. 

To control for factors that may disturb the estimation, we added year-fixed effects and country-fixed effects. In column (2) of Table 2, we added country-pair fixed effects to control for all potential factors between countries. In this regression, the impact of PTAs on bilateral trade is again positive and significant at 1 % level. However, we found that the marginal effect of PTA on bilateral trade dropped from 0.165 (as shown in column 1) to 0.038 (as shown in column 2), which means that when we control for more paired country factors, the marginal effect decreases. In addition, we found no change in the influence of exporter/importer countries income on bilateral trade flow as the coefficient impact is positive. 

We further added the country-year fixed effects to control for the country-year level factors, which may disturb the estimation. The result in column (3), shows that the marginal effect of PTA on bilateral trade decreases further to 0.0356 while it is still significant at the 1% level. The marginal effect given by the structural gravity model indicates that the bilateral trade would increase by 3.56% if countries have RTA between them. Additionally, we investigated the impact of PTAs with the same currency holding other parameters constant. Results showed that PTAs increase bilateral trade flow if these countries have the same currency. Furthermore, a common currency is good for decreasing exchange rate volatility to zero [79, 80]. 

For the first three columns, we used the PTA dummies listed in the WTO. However, Baier et al. stated that dummy variables hardly capture the heterogeneous effect of PTAs [70]; hence, the marginal effect was not accurate. Therefore, we constructed the coverage index of PTAs. The results of the coverage PTAs index reported in column (4)–(6) of Table 2. The results showed that the impact of the coverage index is positive and significant at the 1% level. This implies that under the global production network background, the product-based coverage of PTAs can promote bilateral trade significantly. Similarly, others parameters promote bilateral trade. We tested the coverage index of PTAs impact on bilateral trade with economic scale holding others parameters constant. The results again validate the promoting effect of the coverage index on bilateral trade. The findings in the column (4) indicated that the coverage index of PTAs has a positive impact on bilateral trade. The same currency promote trade between the member countries. In conclusion, the baseline results of the structural gravity model showed that the bilateral trade would increase if two countries have a common trade agreement and the trade promotion effect is positively affected by the product-based coverage of PTAs suggesting that market access openness of PTAs can significantly promote bilateral trade, supporting the trade creation effect of trade agreement.

Table 2.

Empirical results of how PTAs affects bilateral trade

Variables MD

（1） MD

（2） MD

（3） MD

（4） MD

（5） MD

（6）

PTA 0.1527*** 0.0375*** 0.0350*** 

 (0.024) (0.007) (0.005) 

PTA_Ratio 0.5711*** 0.0320** 0.0413***

 (0.071) (0.014) (0.010)

Lnyi 0.3111*** 0.2946*** 0.3133*** 0.2941*** 

 (0.004) (0.004) (0.004) (0.004) 

Lnyj 0.3585*** 0.3498*** 0.3607*** 0.3493*** 

 (0.005) (0.005) (0.005) (0.005) 

LnDis -0.9903*** -0.9876*** 

 (0.017) (0.017) 

Contig 0.3975*** 0.4067*** 

 (0.091) (0.091) 

Comlang 0.1230*** 0.1195*** 

 (0.028) (0.028) 

Colony 1.0761*** 1.0821*** 

 (0.093) (0.093) 

Comcur 2.5615*** 0.0782*** 0.0346*** 2.5605*** 0.0764*** 0.0313***

 (0.147) (0.010) (0.009) (0.147) (0.010) (0.009)

Year FE Y Y Y Y 

Country FE Y Y 

Country-pair FE Y Y Y Y

Country-year FE Y Y

Obs 721,149 721,135 721,135 721,149 721,135 721,135

R2 0.8273 0.9896 0.9965 0.8274 0.9896 0.9965

Note: *, **, and *** represent significance at the 10%, 5%, and 1% levels, respectively. The t-statistic of the robust standard deviation of the estimated coefficients is given in parentheses. The standard deviation is clustered at the country-pair level. MD= model

4.2 Robustness checks 

Scholars have constructed other variables to measure the heterogeneity of PTAs according to the content of PTAs [22–24]. To distinguish our product-based coverage index of PTAs from their measures, we added these variables in the robustness checks. We first compared the product-based coverage index of PTAs with the depth and flexibility measures, and the results are shown in Table 3. We can conclude that the product-based coverage index of PTAs is positively correlated with the depth and flexibility of PTAs, with correlation coefficients of 0.327 and 0.65, respectively. The results show that there are differences between the product-based coverage index and the depth index (depth_rasch) in measuring the coverage of PTAs. The product-based coverage index mainly focuses on the products mentioned in the PTAs, while the depth index covers the seven key provisions. The correlation between the product-based coverage index and flexibility of PTAs (flexescape) is 0.65, suggesting that PTAs are more flexible if they cover more products. We checked the variance inflation factor (VIF) to make sure that there is no multicollinearity problem. The results in the last column of Table 3 shows that there is no multicollinearity problem with the VIF values of the three variables below 10.

Table 3.

Correlation matrix of different measures of PTAs 

 PTA_ratio depth_rasch flexescape VIF

PTA_ratio 1.000 2.35

depth_rasch 0.327*** 1.000 2.12

flexescape 0.650*** -0.093*** 1.000 1.37

Note: We calculated the correlation matrix using Pearson’s method. *, **, and *** represent 10%, 5%, and 1% significance levels, respectively. We use the bilateral trade variable as the independent variable and three measures of PTAs as the dependent variable to calculate the VIF value, which is shown in the last column.

We then added these measures of PTAs in our robustness checks to test the effect of the product-based coverage index of PTAs on bilateral trade. The empirical results in column (1) of Table 4 show that the coverage index of PTAs has a significant positive effect on bilateral trade. Column (1) shows the robustness check of the baseline models, with the depth and flexibility measures of PTAs controlled. We also find that the depth of PTAs can promote bilateral trade, but the marginal effect (0.0316) is less than the coverage index (0.0450). The flexibility of PTAs negatively affects bilateral trade, because the PTAs are flexible to adjust their policies for other purposes without violating the terms of an agreement. They could not cut the tariff as promised because they could withdraw the PTAs. This is important for countries to design flexible PTAs. Flexible PTAs provide for legally accepted opt-outs without leading to a de jure breach of an agreement and encompass exit options, duration and renegotiation clauses, reservations, escape clauses, and withdrawal clauses, and hence help members to cooperate better [23]. The product-based coverage of PTAs differs them in focusing on the market access openness of PTAs instead of other aspects.

8: Table 4. Explain why you decided to do this? I find the endeavor interesting and worthwhile, but it comes across as a bit ad-hoc. Provide some rationale. Also, maybe you don’t want to call it “robustness” checks. I think really what you are doing is showing how your measure is different from and complements depth and flexibility, frame it as such.

Response: Again, we very much appreciate your kind work. In this version, we take you advice and frame table 4 as robustness checks. 

9: Your note for table 5 just says “see note under Table 3. I don’t think the notes are or should be the same

Response: Again, we very much appreciate your kind work. In this version, we write notes for each table in detail.

---

## [Editor Report · Decision Letter 3]

12 Mar 2021

The impact of Preferential Trade Agreements on Bilateral Trade: A Structural Gravity Model Analysis

PONE-D-20-21416R3

Dear Dr. zhang,

We’re pleased to inform you that your manuscript has been judged scientifically suitable for publication and will be formally accepted for publication once it meets all outstanding technical requirements.

Kind regards,

Bernhard Reinsberg, Ph.D

Academic Editor

PLOS ONE
---

## [Editor Report · Acceptance letter]

19 Mar 2021

PONE-D-20-21416R3 

The impact of Preferential Trade Agreements on Bilateral Trade: A Structural Gravity Model Analysis 

Dear Dr. Zhang:

I'm pleased to inform you that your manuscript has been deemed suitable for publication in PLOS ONE. Congratulations! Your manuscript is now with our production department. 

Kind regards, 

on behalf of

Dr. Bernhard Reinsberg 

Academic Editor

PLOS ONE